# Embodied Scene-aware Human Pose Estimation

**Zhengyi Luo**[1] *   **Shun Iwase** [1]*   **Ye Yuan**[1]   **Kris Kitani**[1]

[1] Carnegie Mellon University

https://zhengyiluo.github.io/projects/embodied_pose/

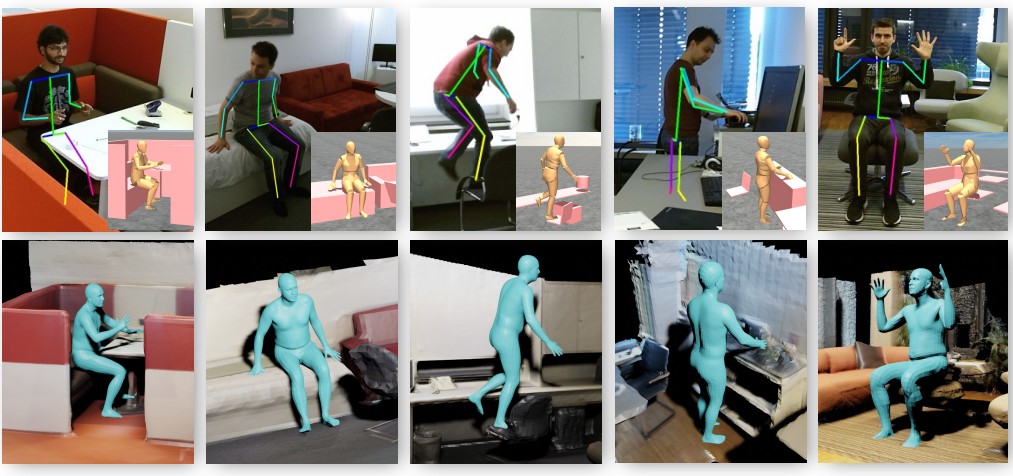

Figure 1: From a monocular video input in a known scene, we estimate 3D absolute body pose by controlling a simulated character to match 2D observation in a sequential manner.

## Abstract

We propose *embodied scene-aware human pose estimation* where we estimate 3D poses based on a simulated agent's proprioception and scene awareness, along with external third-person observations. Unlike prior methods that often resort to multistage optimization, non-causal inference, and complex contact modeling to estimate human pose and human scene interactions, our method is one-stage, causal, and recovers global 3D human poses in a simulated environment. Since 2D third-person observations are coupled with the camera pose, we propose to disentangle the camera pose and use a multi-step projection gradient defined in the global coordinate frame as the movement cue for our embodied agent. Leveraging a physics simulation and prescanned scenes (*e.g.,* 3D mesh), we simulate our agent in everyday environments (library, office, bedroom, *etc.*) and equip our agent with environmental sensors to intelligently navigate and interact with the geometries of the scene. Our method also relies only on 2D keypoints and can be trained on synthetic datasets derived from popular human motion databases. To evaluate, we use the popular H36M and PROX datasets and achieve high quality pose estimation on the challenging PROX dataset without ever using PROX motion sequences for training. Code and videos are available on the project page.

## 1   Introduction

We consider a 3D human pose estimation setting where a calibrated camera is situated within a prescanned environment (with known 3D mesh geometries). Such a setting has a wide range

---

*Equal Contribution.

36th Conference on Neural Information Processing Systems (NeurIPS 2022).

of applications in mixed reality telepresence, animation, and gaming, at home robotics, video surveillance, *etc.*, where a single fixed RGB camera is often used to observe human movement. However, even under such constraints, severe occlusion, low-textured garments, and human-scene interactions often cause pose estimation methods to fail catastrophically. Especially when estimating the global 3D human pose, state-of-the-art methods often resort to expensive multistage batch optimization [6], motion prior or infilling [31, 48], and depth information to recover reasonable global translation and orientation. Consequently, these methods have limited use cases in time-critical applications. Multistage optimization can also suffer from failure of convergence, where the optimizer cannot find a good solution due to occlusion, numerical instability, or poor initialization.

The effective use of scene information is also critical for human pose estimation. Unfortunately, the lack of datasets with paired ground-truth 3D body pose and annotated contact labels makes it difficult to incorporate scene information to estimate pose. Thus, popular methods often resort to 1) minimizing the distance between predefined contact points in the body and the scene [6], 2) using vertex-based PROXimity [49] as a proxy scene representation, and 3) semantic labels as affordance cues [17]. These methods, while effective in modeling human-scene interactions, are expensive to compute and heavily rely on a batch optimization procedure to achieve good performance. More complex scenes are also harder to model in a batch optimization framework, as the number of contact constraints increases significantly as the number of geometries increases.

We hypothesize that **embodiment** may be a key element in estimating accurate absolute 3D human poses. Here, we define embodiment as having a tangible existence in a simulated environment in which the agent can act and perceive. In particular, humans move according to their current body joint configuration (proprioception), movement destination (goal), and an understanding of their environment's physical laws (scene awareness and embodiment). Under these guiding principles, we propose **embodied scene-aware human pose estimation** as an alternative to optimization and regression-based methods. Concretely, we estimate poses by controlling a physically simulated agent equipped with environmental sensors to mimic observations from third-person in a **streaming** fashion. At each timestep, our agent receives movement goals from a third-person observation and computes its torque actuation based on (1) current agent state: body pose, velocities, *etc.*, (2) third-person observation, and (3) environmental features from the agent's point of view. Acting in a physics simulator, our agent can receive timely feedback about physical plausibility and contact to compute actions for the next time frame. Although there exist prior methods that also use simulated characters, we draw a distinction in how the next frame pose is computed. Previous work first regress the kinematic pose [4, 46] without considering egocentric features such as body and environmental states. As a result, when the kinematic pose estimation methods fail due to occlusion, these simulated agents will receive unreasonable target poses. Our method, on the other hand, predicts target poses by comparing the current body pose with third-person observations. When third-person observations are unreliable, the target poses will be extrapolated from environmental cues and current body states. Thus, our method is robust to occlusion and can recover from consecutive frames of missing body observations. Our method is also capable of estimating pose for long-term (several minutes to perpetual), in sub-real-time ($\sim$ 10 fps), while creating physically realistic human scene interactions.

To achieve this, we propose a multi-step projection gradient (MPG) to provide movement cues to our embodied agent from 2D observations. At each time step, MPG performs iterative gradient descent starting from the current pose of the humanoid to exploit the temporal coherence of human movement and better avoid local minima. MPG also directly provides temporal body and root movement signals to our agent in the world coordinate frame. Thus, our networks does not need to take the 3D camera pose as input explicitly. To control a simulated character, we extend the Universal Humanoid Controller (UHC) and the kinematic policy proposed in previous work [21] to support the control of humanoids with different body shapes in environments with diverse geometries. Finally, we equip our agent with an occupancy-based environmental sensor to learn about affordances and avoidance. Since only 2D keypoint coordinates are needed by our method to estimate body motion, our model can be learned from purely synthetic data derived from popular motion databases.

To summarize, our contribution is as follows: 1) we propose embodied scene-aware human pose estimation, where we control a simulated scene-aware humanoid to mimic visual observations and recover accurate global 3D human pose and human-scene interactions; 2) we formulate a multi-step projection gradient (MPG) to provide a camera-agnostic movement signal for an embodied agent to match 2D observations; and 3) we achieve state-of-the-art result on the challenging PROX dataset while our method remains sub-real-time, causal, and trained only on synthetic data.

## 2 Related Work

**Local and Absolute 3D Human Pose Estimation.** It is popular to estimate the 3D human pose in the local coordinate system (relative to the root) [1, 11, 12, 13, 14, 15, 16, 20, 24, 25, 27, 30, 37, 39] where the translation is discarded and the rotation is defined in the camera coordinate frame. Among these methods, some focus on lifting 2D keypoints to 3D [1, 27], while others use a template 3D human mesh (the SMPL [19] body model and its extensions [26, 32]) and jointly recover body shape and joint angles. Among these methods, some [15, 37, 46] make use of iterative optimization to better fit 2D keypoints. SPIN [15] devises an optimization-in-the-loop fitting procedure to provide a better starting point for a pose regressor. NeuralGD [37] and SimPOE [46] both utilize a learned gradient update to iteratively refine body pose and shape, where optimization starts from either zero pose (NeuralGD) or estimated kinematic pose (SimPOE). Compared to these methods, our MPG starts the computation process from the previous time-step pose to leverage the temporal coherence in human motion. While estimating pose in the local coordinate frame is useful in certain applications, it is often desirable to recover the pose in the absolute/global space to fully leverage the estimated pose for animation or telepresence. However, recovering the absolute 3D body pose that includes root translation is challenging due to limb articulation, occlusion, and depth ambiguity. Among the work that attempt to estimate the absolute human pose [24, 27, 30, 33, 43], most take a two-stage approach where the root-relative body pose is first regressed from a single image and then a root translation is estimated from image-level features. Compared to these methods, we leverage physical laws and motion cues to autoregressively recover both root movement and body pose.

**Scene-aware Human Pose and Motion Modeling.** The growing interest in jointly modeling human pose and human-scene interactions has led to a growing popularity in datasets containing both human and scenes [7, 23, 41]. In this work, we focus on the PROX [7] dataset, where a single human is recorded moving in daily indoor environments such as offices, libraries, bedrooms, etc. Due to the diversity of motion and human-scene interaction (sitting on chairs, lying on beds, etc.) and lack of ground-truth training data, the PROX dataset remains highly challenging. Consequently, state-of-the-art methods are mainly optimization-based [7, 31, 47] while leveraging the full set of observations (scene scans, camera pose, depth). HuMoR [31] and LEMO [47] achieves impressive result by using large-scale human motion datasets to learn strong prior of human movement to constrain or infill the missing human motion. However, multistage optimization can often become unstable due to occlusion, failures in 2D keypoint detections, or erroneous depth segmentation.

**Physics-based humanoid control for pose estimation.** Physical laws guard every aspect of human motion and there is a growing interest in enforcing physics prior in human pose estimation [4, 9, 21, 35, 36, 42, 44, 45, 46]. Some methods prefer to use physics as a postprocessing step and refine initial kinematic pose estimates [35, 36, 42], while ours follows the line of work where a simulated character is controlled to estimate pose inside a physics simulation [4, 9, 21, 46]. Although both approaches have their respective strong suites, utilizing a physics simulation simplifies the process of incorporating unknown scene geometries and can empower real-time pose estimation applications. Among the simulated character control-based methods, Neural Mocon [9] utilizes sampling-based control and relies on the learned motion distribution from the training split of the dataset to acquire quality motion samples. SimPOE [46] first extracts the kinematic pose using off-the-shelf method [13] and then uses a simulated agent for imitation. PoseTriplet [4] proposes using a self-supervised framework to co-evolve a pose estimator, imitator, and hallucinator. Compared to ours, SimPoE and PoseTriplet both rely on a kinematic pose estimator as backbones [13, 27] and require well-observed 2D keypoints for reliable pose estimates. They also require lengthy policy training time and do not model any scene geometries. Our work extends Kin_poly [21], where a simulated character is used to perform *first-person* pose estimation. While Kin_poly uses egocentric motion as the only motion cue, we propose a Multistep Projection Gradient to leverage thrid-person observations. Compared to their object-pose based scene feature, we use an occupancy-based scene feature inspired by the graphics community [5, 38] to better model scene geometries.

## 3 Method

Given a monocular video $I_{1:T}$ taken using a calibrated camera in a known environment, our aim is to estimate, track, and simulate 3D human motion in metric space. We denote the 3D human

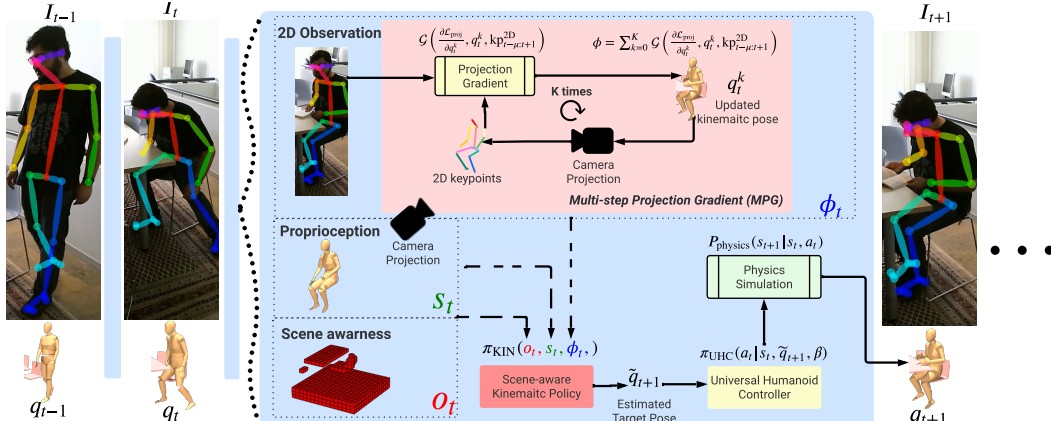

Figure 2: Our embodied human pose estimation framework: given 2D keypoint features, our model computes the next-time frame pose based on third-person 2D observation (multi-step projection gradient $\phi_t$), proprioception features (body state $s_t$), and scene (occupancy $o_t$).

pose as $q_t \triangleq (r_t^{\mathrm{R}}, r_t^{\mathrm{T}}, \theta_t)$, consisting of the 3D root position $r_t^{\mathrm{T}}$, the orientation $r_t^{\mathrm{R}}$ and the angles of the body joints $\theta_t$. Combining 3D pose and velocity, we have the character's state $s_t \triangleq (q_t, \dot{q}_t)$. As a notation convention, we use $\tilde{\cdot}$ to represent kinematic poses (without physics simulation), $\hat{\cdot}$ to denote ground-truth quantities from Motion Capture (MoCap), and normal symbols without accents for values from the physics simulation. In Sec 3.1, we will first summarize our extension to the Universal Humanoid Controller [21] to support controlling people with different body proportions. Then, in Sec.3.2, we will describe the state space of our character's control policy: the multi-step projection gradient (MPG) as a feature and occupancy based scene feature. Finally, we will detail the training procedure to learn our embodied pose estimator using synthetic datasets in Sec 3.3. Our pose estimation pipeline is visualized at Fig.2.

## 3.1 Preliminaries: Physics-based Character and Universal Humanoid Controller (UHC)

To simulate people with varying body shapes, we first use an automatic simulated character creation process similar to SimPoE [46] to generate human bodies with realistic joint limits, body proportions and other physical properties based on the popular human mesh model SMPL [19]. In SMPL, body joints are recovered from mesh vertices and a joint regressor, so the length of each limb may vary according to the body pose. This is undesirable as varying the length of the body limbs is physically inaccurate. Recovering body joints by first computing a large number of vertices (6890) followed by a joint regressor is also time-consuming. Therefore, we fix the limb lengths using those calculated from the T pose (setting the joint angles to zero) and use forward kinematics to recover the 3D location of the body joints $J_t$ from joint angles. Given the shape of the body $\beta$ and the pose $q_t$, we define the forward kinematics function as $\mathrm{FK}_\beta(q_t) \to J_t$.

To learn a universal humanoid controller that can perform everyday human motion with different body types, we follow the standard formulation in physics-based character and model controlling a humanoid for motion imitation as a Markov Decision Process (MDP) defined as a tuple $\mathcal{M} = \langle \mathcal{S}, \mathcal{A}, \mathcal{T}, R, \gamma \rangle$ of states, actions, transition dynamics, reward function, and discount factor. The state $s \in \mathcal{S}$, reward $r \in \mathcal{R}$, and the transition dynamics $\mathcal{T}$ are determined by the physics simulator, while the action $a \in \mathcal{A}$ is given by the control policy $\pi_{\mathrm{UHC}}(a_t | s_t, \hat{q}_{t+1}, \beta)$. Here we extend the Universal Humanoid Controller proposed in Kin_poly [21] and condition the control policy on the current simulation state $s_t$, the reference motion $\hat{q}_t$, and the body shape $\beta$. We employ Proximal Policy Optimization (PPO) [34] to find the optimal control policy $\pi_{\mathrm{UHC}}$ to maximize the expected discounted reward $\mathbb{E}\left[\sum_{t=1}^{T} \gamma^{t-1} r_t\right]$. For more information on the UHC network architecture and training procedure, please refer to Kin_poly [21] and our Appendix.

## 3.2 Embodied Scene-Aware Human Pose Estimation

To estimate the global body pose from the video observations, we control a simulated character to follow the 2D keypoint observations in a sequential decision manner. Assuming a known camera

pose and scene geometry, we first use an off-the-shelf 2D keypoint pose estimator, DEKR [3], to extract 2D keypoints $\text{kp}_t^{2D}$ at a per-frame level. Then, we use HuMoR [31] to extract the first-frame 3D pose $\widetilde{q}_0$ and body shape $\beta$ from the RGB observations as initialization. Given $\widetilde{q}_0$ and $\beta$, we initialize our simulated character inside the pre-scanned scene, and then we attempt to estimate poses which match observed 2D keypoints in a streaming fashion.

**Multi-step Projection Gradient (MPG) as a Motion Feature.** To match the motion of the 3D simulated humanoids to 2D keypoint observations, it is essential to provide 3D information about how poses should be modified in the global coordinate frame (coordinate frame of the scene geometry and the 3D humanoid). As 2D keypoints are coupled with camera pose (the same body pose corresponds to different 2D keypoints based on the camera pose), it is problematic to directly input 2D observations without a proper coordinate transform. To avoid this issue, prior arts that use simulated character control [4, 46] often estimate the kinematic human pose in the camera coordinate frame using a kinematic pose estimator. However, conducting physics simulation using the camera coordinate frame is challenging, since the up-vector of the camera is not always aligned with the direction of gravity. As a result, manual alignment of the character's up-vector to the gravity direction is often needed for a successful simulation [4].

To decouple the pose of the camera from the pose estimation pipeline, we propose using a multi-step projection gradient (MPG) feature as an alternative to estimated kinematic pose as an input to the policy. Intuitively, the MPG is a modified (learned) sum of 2D projection loss gradients that encodes how much the 3D pose should be adjusted for to fit the next frame 2D keypoints. Given the current global state of the humanoid $q_t$ and the camera projection function $\pi_{\text{cam}}$, we define MPG $\phi$ as:

$$\phi_t = q_t^K - q_t^0 = \sum_{k=0}^{K} \mathcal{G}(\frac{\partial \mathcal{L}_{\text{proj}}}{\partial q_t^k}, q_t^k, \text{kp}_{t-\mu:t+1}^{2D}), q_t^{k+1} = q_t^k - s \cdot \mathcal{G}(\frac{\partial \mathcal{L}_{\text{proj}}}{\partial q_t^k}, q_t^k, \text{kp}_{t-\mu:t+1}^{2D}) \quad (1)$$

and $\mathcal{L}_{\text{proj}} = \|\pi_{\text{cam}}(\text{FK}_\beta(q_t)) - \text{kp}_{t+1}^{2D}\|_2$ is the $L_2$ reprojection loss with respect to the observed 2D keypoints $\text{kp}_{t+1}^{2D}$. $s$ is the step size of the gradient and $k$ indexes into the total number of gradient descent steps $K$. The 2D project loss gradient $\frac{\partial \mathcal{L}}{\partial q_t}$ represents how the root pose and body joints should change to better match the 2D key points of the next frame and contains rich geometric information independent of the camera projection function $\pi_{\text{cam}}$.

Due to noise in the detection, occlusion, and depth ambiguity of 2D key points, a single-step instantaneous projection gradient $\frac{\partial \mathcal{L}_{\text{proj}}}{\partial q_t^k}$ can be erroneous and noisy, especially the gradient of root translation and orientation. Thus, we augment $\frac{\partial \mathcal{L}_{\text{proj}}}{\partial q_t^k}$ with a geometric and learning-based transform $\mathcal{G}$ that is intended to remove noise. In this way, the MPG feature is an aggregation of a collection of modified gradients to compute a more stable estimate of the 2D projection loss gradient.

To better recover root rotation, we use a temporal convolution model (TCN) similar to VideoPose3D [27] and PoseTriplet [4] to regress camera-space root orientation from 2D keypoints. We notice that while TCN may produce an unreasonable pose estimate during server occlusion, its root orientation estimation remains stable. We compute the camera space root orientation using a causal TCN $\mathcal{F}_{\text{TCN}}$ and transform it into the world coordinate system $r_t'^{\text{R}}$ using the inverse camera rotation: $r_t'^{\text{R}} = \pi_{\text{cam}}^{-1}(\mathcal{F}_{\text{TCN}}(\text{kp}_{t-\mu:t+1}^{2D}))$. We then calculate the difference between the current root rotation $r_t^{\text{R}}$ and $r_t'^{\text{R}}$ as an additional movement gradient signal. Given the current 3D pose $q_t^k$ and the projection function $\pi_{\text{cam}}$, we can obtain a better root translation $r_t^T$ by treating the 3D keypoints obtained as a fixed rigid body and solving for translation using linear least squares. Combining the $\mathcal{F}_{\text{TCN}}$ model and the geometric transformation step, we obtain the function $\mathcal{G}$ to obtain a better root translation and orientation gradient. By taking multiple steps, we avoid getting stuck in local minima and obtain better gradient estimates. For a more complete formulation of MPG, see our Appendix.

**Scene Representation.** To provide our embodied agent with an understanding of its environment, we provide our policy with two levels of scene-awareness: 1) we simulate the geometry of the scene inside a physics simulation, which implicitly provides contact and physics constraints for our policy. As the scanned meshes provided by the PROX dataset often have incorrect volumes, we use a semi-automatic geometric fitting procedure to approximate the scene by simple shapes such as rectangles and cylinders. The modified scene can be visualized together in Eq.2. For more

**ALGORITHM 1:** Learning embodied pose estimator via dynamics-regulated training.

---

**Input:** Pre-trained controller $\boldsymbol{\pi}_{\text{UHC}}$, ground truth motion dataset $\widehat{\boldsymbol{Q}}$ ;
**while** *not converged* **do**
    $M_{\text{dyna}} \leftarrow \emptyset$ initialize sampling memory ;
    **while** $M_{dyna}$ *not full* **do**
        $\boldsymbol{q}_{1:T} \leftarrow$ sample random sequence of human pose ;
        $\text{kp}_{1:T}^{2D} \leftarrow \pi_{\text{cam}}(\boldsymbol{q}_{1:T})$                   // compute paired 2D projection. ;
        **for** $t \leftarrow 1...T$ **do**
            $\boldsymbol{s}_t \leftarrow (\boldsymbol{q}_t, \dot{\boldsymbol{q}}_t)$
            $\phi_t \leftarrow \sum_{k=0}^{K} \mathcal{G}(\frac{\partial \mathcal{L}_{\text{proj}}}{\partial \boldsymbol{q}_t^k}, \boldsymbol{q}_t^k, \text{kp}_{t-\mu:t+1}^{2D})$       // movement cues from 2D observations. ;
            $\widetilde{\boldsymbol{q}}_{t+1} \leftarrow \boldsymbol{\pi}_{\text{KIN}}(\boldsymbol{s}_t, \boldsymbol{\phi}_t, \boldsymbol{o}_t) + \boldsymbol{q}_t$
            $\boldsymbol{s}_{t+1} \leftarrow \mathcal{T}(\boldsymbol{s}_{t+1}|\boldsymbol{s}_t, \boldsymbol{a}_t), \boldsymbol{a}_t \leftarrow \boldsymbol{\pi}_{\text{UHC}}(\boldsymbol{a}_t | \boldsymbol{s}_t, \widetilde{\boldsymbol{q}}_{t+1})$ // physics simulation and agent control using $\boldsymbol{\pi}_{\text{UHC}}$
            store $(\boldsymbol{s}_t, \boldsymbol{a}_t, \boldsymbol{r}_t, \boldsymbol{s}_{t+1}, \widehat{\boldsymbol{q}}_t, \widetilde{\boldsymbol{q}}_{t+1})$ into memory $M_{\text{dyna}}$
        **end**
    **end**
**end**
$\boldsymbol{\pi}_{\text{KIN}} \leftarrow$ Supervised learning update using experiences collected in $M_{\text{dyna}}$ for 10 epoches.

---

information about our scene simplification process, please refer to the Appendix. 2) We use a voxel-based occupancy map to indicate the location and size of the scene geometries, similar to [5, 38]. Occupancy provides basic-level scene and object awareness to our agent and is robust to changing scene geometries. Concretely, we create a cube of 1.8 meters edge length as the sensing range of our agent's surroundings and create a 16 by 16 by 16 evenly spaced grid. At the center of each point on the grid, we sample the signed distance function of the scene $\mathcal{F}_s$ to obtain how far the sampling point is from the geometries of the scene. Then, we threshold the gird feature to provide a finer scene information. In time step $t$, for threshold $\sigma = 0.05$, the value of point $p_i$ in the grid can be computed as $o_t^i$, and $\boldsymbol{o}_t = \{o_0^i, o_1^i, \ldots, \}$:

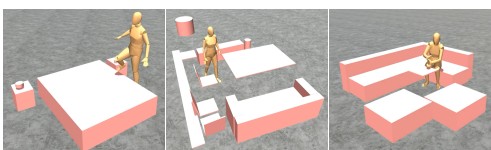

$$o_t^i = \begin{cases} 1 & \mathcal{F}_s(p_i) \leq 0 \\ 0 & \mathcal{F}_s(p_i) > \sigma \quad (2) \\ 1 - \mathcal{F}_s(p_i)/\sigma & \text{otherwise} \end{cases}$$

Figure 3: Our modified scene for simulation.

**Scene-aware Kinematic Policy.** To train and control our embodied agent in the physics simulator, we extend a policy network similar to Kin_poly [21], with the important distinction that our networks perform third-person pose estimation in a real-world scene. Under the guiding principle of embodied human pose estimation, our kinematic policy $\boldsymbol{\pi}_{\text{KIN}}$ computes a residual 3D movement $\dot{\tilde{\boldsymbol{q}}}_t$ based on proprioception $\boldsymbol{s}_t$, movement goal $\boldsymbol{\phi}_t$, and scene awareness $\boldsymbol{o}_t$:

$$\dot{\tilde{\boldsymbol{q}}}_t = \boldsymbol{\pi}_{\text{KIN}}(\boldsymbol{s}_t, \boldsymbol{\phi}_t, \boldsymbol{o}_t), \quad \tilde{\boldsymbol{q}}_{t+1} = \dot{\tilde{\boldsymbol{q}}}_t + \tilde{\boldsymbol{q}}_t. \tag{3}$$

Specifically, the input $\boldsymbol{q}_t$ contains the current body joint configurations, $\boldsymbol{\phi}_t$ provides the movement signal per step, while $\boldsymbol{o}_t$ provides awareness to the agent's surroundings. Within $\boldsymbol{\pi}_{\text{KIN}}$, an agent-centric transform $T_{\text{AC}}$ is used to transform $\boldsymbol{q}_t$, $\boldsymbol{\phi}_t$ and $\boldsymbol{s}_t$ into the agent-centirc coordinate system. The output $\dot{\tilde{\boldsymbol{q}}}_t$, is defined as $\dot{\tilde{\boldsymbol{q}}}_t \triangleq (\dot{\tilde{\boldsymbol{r}}}_t^{\text{R}}, \dot{\tilde{\boldsymbol{r}}}_t^{\text{T}}, \dot{\tilde{\boldsymbol{\theta}}}_t)$, which computes the velocity in root translation $\dot{\tilde{\boldsymbol{r}}}_t^{\text{R}}$ and orientation $\dot{\tilde{\boldsymbol{r}}}_t^{\text{T}}$, as well as the joint angles $\dot{\tilde{\boldsymbol{\theta}}}_t$. Notice that $\dot{\tilde{\boldsymbol{q}}}_t$ computes the residual body movement instead of the full kinematic body pose $\tilde{\boldsymbol{q}}_{t+1}$ and establishes a better connection with the movement feature $\boldsymbol{\phi}_t$. During inference, $\boldsymbol{\pi}_{\text{KIN}}$ works in tandem with $\boldsymbol{\pi}_{\text{UHC}}$ and computes the next-step target pose $\tilde{\boldsymbol{q}}_{t+1}$ for the humanoid controller to mimic. In turn, $\boldsymbol{\pi}_{\text{KIN}}$ receives timely feedback on physical realism and contact information based on the simulated agent state.

### 3.3 Learning the embodied human pose estimator

Since $\boldsymbol{\pi}_{\text{KIN}}$ relies only on 2D keypoints, we can leverage large-scale human motion databases and train our pose estimation network using synthetic data. We use the AMASS [22], H36M [10], and

Kin_poly [21] datasets to dynamically generate 2D keypoints and 3D pose pairs for training. At the beginning of each episode, we sample a motion sequence and randomly change its direction and starting position in a valid range, similar to the procedure defined in NeuralGD [37]. Then we use a randomly selected camera projection function $\pi_{\text{cam}}$ to project 3D keypoints onto 2D images to obtain paired 2D keypoints and human motion sequences. During training, we initialize the agent with the ground-truth 3D location and pose. The H36M and Kin_poly dataset contains simple human-scene interactions, such as sitting on chairs and stepping on boxes. To better learn object affordance and collision avoidance, we also randomly sample scenes from the PROX dataset and pair them with motions from the AMASS dataset. Although these motion sequences do not correspond to the sampled scene, the embodied agent will hit the scene geometries while trying to follow the 2D observations. We stop the simulation when the agent deviates too far away from the ground-truth motion to learn scene affordance and avoidance. For more details about our synthetic data sampling procedure, please refer to the Appendix.

We train the kinematic policy as a per-step model together with physics simulation, following the dynamics-regulated training procedure in [21] (Alg. 1.). Using this procedure, we can leverage the pre-learned motor skills from the UHC and let the kinematic policy act directly in a simulated physical space to obtain feedback on physical plausibility. In each episode, we connect the UHC and the kinematic policy using the result of the physics simulation $q_{t+1}$ to compute the input features to the kinematic policy. Using $q_{t+1}$ verifies that $\widetilde{q}_t$ can be followed by our UHC and informs our $\pi_{\text{KIN}}$ of the current humanoid state to encourage the policy to adjust its predictions to improve stability. Here, we only train our $\pi_{\text{KIN}}$ using supervised learning, as third-person pose estimation is a more well-posed problem and requires less exploration than estimating pose from only egocentric videos [21]. For each motion sequence, our loss function is defined as $L1$ distance between the predicted kinematic pose and the ground truth plus a prior loss using a learned motion transition prior $p_\theta$ and the encoder $q_\phi$ from HuMoR [31]. The prior term encourages the generated motion to be smoother:

$$\mathcal{L} = \sum_{t=0}^{T} w_{\text{L1}} \|\hat{q}_t - \tilde{q}_t\|_1 + w_{\text{prior}} D_{\text{KL}}(q_\phi(z_t|\tilde{q}_t, \tilde{q}_{t-1}) || p_\theta(z_t|\tilde{q}_{t-1})), \qquad (4)$$

## 4 Experiments

**Implementation details.** We use the MuJoCo [40] physics simulator and use modified scene geometries as environments. The training process or our embodied pose estimator takes around 2 days on a RTX 3090 with 30 CPU threads. During inference, our network is causal and runs at ∼10 FPS on an Intel desktop CPU (not counting the time for 2D keypoint pose estimation, which runs at around 30 FPS [3]). For more details on the implementation, please refer to the Appendix.

**Datasets.** Our Universal Humanoid Controller is trained on the AMASS dataset training split that contains high-quality SMPL parameters of 11402 sequences curated from the various MoCap datasets. For training our pose estimator, we use motions from the AMASS training split, the Kin_poly dataset, and H36M dataset. The Kin_poly dataset contains 266 motion sequences of sitting, stepping, pushing, and avoiding, and the H36M dataset consists of 210 indoor videos with 51 different unique actions. Note that we only use the motion sequences from these datasets and not the paired videos. For evaluation, we use videos from the PROX [7] dataset and the test split of H36M.

**Metrics and Baselines.** We compare with a series of state-of-the-art methods HyperIK [16], Me-TRAbs [33], VIBE [13], and SimPOE [46] in the H36M dataset. On the PROX dataset, we compare with the HuMoR [31], LEMO [47], and HyberIK with RootNet [24]. We use the official implementation of the above methods. We use pose-based and plausibility-based metrics for evaluation. To evaluate pose estimation when there is ground-truth MoCap pose, we report the absolute/global and root-relative mean per joint position error: A-MPJPE and MPJPE. On the PROX dataset, since there is no ground truth 3D pose annotation, we report: success rate SR (whether the optimization procedure converges and whether the simulated agents lose balance), one-way average chamfer distance from the point cloud (ACD, acceleration (Accel.c), and scene penetration frequency (Freq.) and distance (Pen.). For a full definition of our evaluation metrics, please refer to the Appendix.

### 4.1 Results

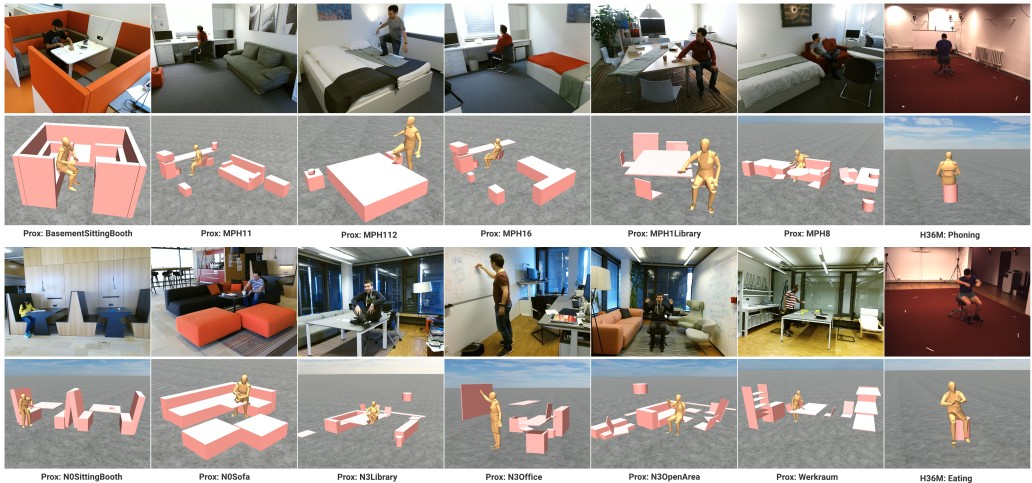

Figure 4: Visualization of our pose estimation result on the PROX and H36M dataset, along with our modified scenes. For each scene, we approximate the scene geometries with simple shapes.

**Results on H36M.** We report the pose estimation on the H36M dataset in Table 1. Here, we can see that our method achieves a reasonable pose estimation result on both root-relative and global pose metrics. Notice that our experiment with the H36M dataset is meant to be a sanity check on our pose estimation framework and is not a fair comparison. Although

Table 1: Our pose estimation result on the H36M dataset. *Since our method does not directly use any image-level feature, uses a smaller set of 2D keypoints for observation, and evaluates on a different set of 3D keypoints, it is not a fair comparison against SOTA methods.

| | Image-features | A-MPJPE ↓ | MPJPE ↓ | PA-MPJPE ↓ |
|---|---|---|---|---|
| MeTRAbs [33] | ✓ | 223.9 | 49.3 | 34.7 |
| VIBE [16] | ✓ | - | 61.3 | 43.1 |
| SimPoE [46] | ✓ | - | 56.7 | 41.6 |
| HybrIK [16] w/ RootNet [24] | ✓ | 135.9 | 65.2 | 43.4 |
| Ours* | ✗ | 284.0 | 105.1 | 72.1 |

our method does not obtain the state-of-the-art result on H36M, we want to point out that 1) our method is trained on purely synthetic data and has never seen detected 2D keypoints from H36M videos, 2) we do not use any image-level information and only relies on a small set (12 in total) of 2D keypoints to infer global pose. Under such conditions, it is especially challenging when the person's movement direction is parallel to the forward direction of the camera (the person is moving along the depth direction in the image plane). We also evaluate a different subset of 3D keypoints, as our use of forward kinematics $FK_\beta$ causes our joint definition to be different from prior work. For more clarification and visualization on the H36M evaluation, please refer to the Appendix and project site.

Table 2: Evaluation of plausibility on the PROX dataset. Since PROX has no ground truth annotation, we employ ACD (one-way Average Chamfer Distance) between the segmented point cloud and predicted body mesh as a proxy metric for pose estimation plausibility, as formulated in HuMoR[31]. Notice that the ACD values correspond to a different number of sequences based on the success rate (out of 60 sequences from PROX). Since all of the metrics can only approximate plausibility, we encourage our readers to refer to our project website for visualization of **all** 60 sequences on PROX of our method. Time(s) is the per-frame average runtime evaluated on RTX 2080 Super. Note that only HybrIK [16] w/ RootNet [24] and ours are not multi-stage optimization based methods (indicated by the coloum "Opti.") and offers a relatively fair comparison.

| | | Phys. | Scene | Opti. | SR ↑ | ACD ↓ | Accel. ↓ | Ground Freq. ↓ | Ground Pen. ↓ | Scene Freq. ↓ | Scene Pen. ↓ | Time(s) ↓ |
|---|---|---|---|---|---|---|---|---|---|---|---|---|
| RGB-D | PROX [7] | ✗ | ✓ | ✓ | 98.3% | 181.1 | 157.4 | 10.6 % | 109.1 | 0.7 | 46.8 % | 73.1 |
| | HuMoR [31] | ✗ | ✗ | ✓ | 100% | 325.5 | 3.4 | 2.4% | 113.6 | 0.4 % | 47.5 | 13.72 |
| | LEMO [47] | ✗ | ✓ | ✓ | 71.7% | 173.2 | 3.0 | 0 % | 14.5 | 0.2% | 22.8 | 75.19 |
| RGB | PROX [7] | ✗ | ✓ | ✓ | 90.0% | >1000 | 381.6 | 4.6 % | >1000 | 0.39 % | 36.7 | 47.64 |
| | HuMoR [31] | ✗ | ✗ | ✓ | 100% | 311.5 | 3.1 | 0.1 % | 38.1 | 0.3% | 36.4 | 11.73 |
| | HybrIK [16] w/ RootNet [24] | ✗ | ✗ | ✗ | - | >1000 | 310.6 | 4.4% | 384.7 | 0.6 % | 48.9 | **0.08** |
| | Ours | ✓ | ✓ | ✗ | 98.3% | **171.7** | **9.4** | **0%** | **24.9** | **0 %** | **16.6** | 0.12 |

**Results on PROX.** Table 2 shows our result in the PROX dataset where we compare against SOTA multi-stage batch optimization based methods: LEMO [47] and HuMoR [31]. We also report the

| Ours (Causal) | HybrIK –RGB (Causal) | PROX - RGB (Multi-stage Optimization) | LEMO - RGBD (Multi-stage Optimization) | HuMoR- RGBD (Multi-stage Optimization) | 2D Keypoints |
|---|---|---|---|---|---|

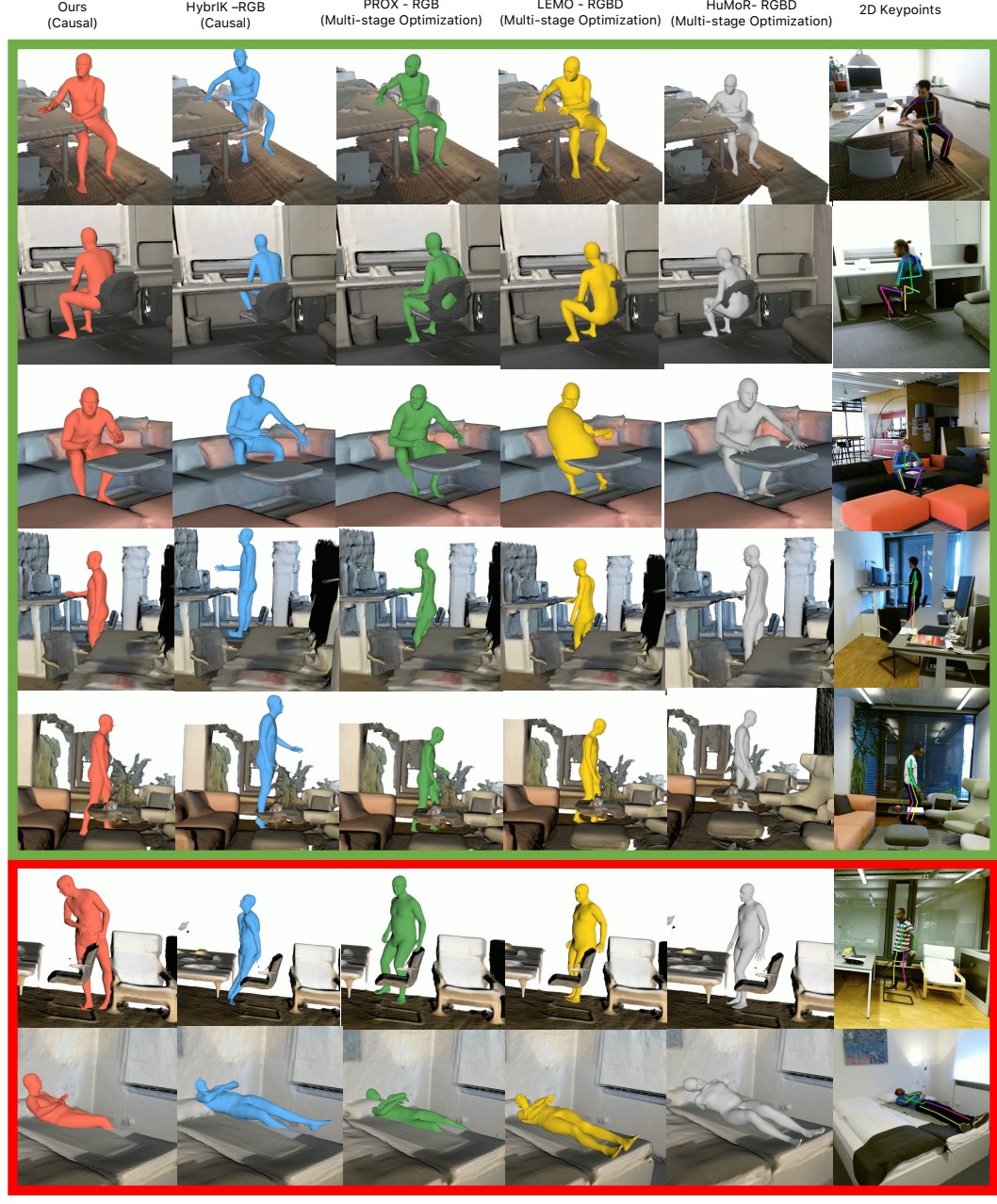

Figure 5: Comparision with SOTA methods. Last two rows are failure cases.

result using HyperIK with RootNet [16] to show that severe occlusion and human scene interaction on the PROX dataset is also challenging to SOTA regression methods. From Table 2, we can see that our method achieves the best ACD (chamfer distance) *without* using any depth information and is competitive in all other metrics against both RGB and RGB-D based methods. Note that LEMO and HuMoR directly optimize the chamfer distance and ground penetration during optimization, while our method relies only on 2D keypoints and laws of physics to achieve plausible pose estimates. Methods with exceedingly high ACD values (> 1000) can be attributed to unstable optimization or unreliable depth estimation, which causes the humanoid to fly away for some sequences.

In Fig.4, we visualize our results in each of the scene from the PROX dataset. Fig.5 compares our result with SOTA methods. We can see that our method can estimate accurate 3D absolute poses that align with the 3D scene. Compared to batch multistage optimization methods that utilize RGB-D input, PROX [7] and LEMO [47], our method can achieve comparable results while running in near-

real-time. Compared to the regression-based method, HybrIK [16] with RootNet [24], our method can estimate much more accurate root translation and body pose. In terms of physical plausibility, optimization-based methods often perform well but can still exhibit floating and penetration. While optimization methods can take over a minute per-frame, catastrophic failure can still occur (as shown in the fourth row). For video quantitative comparison, please refer to our project website.

## 4.2 Ablation

In Table 3, we ablate the main components of our framework. Row 1 (R1) corresponds to not using our modified scene geometry and uses the mesh provided by the PROX dataset; row 2 corresponds to not using any scene awareness (voxel features); R2 uses MPG with only one time-step, which results in nosier gradients; R3 trains a model without using the geometric translation step in $\mathcal{G}$; R4 does not use the TCN model $\mathcal{F}_{\text{TCN}}$ in $\mathcal{G}$; R5 does not leverage modified 3D scenes and uses the original scene scans. R1 shows that our modified geometry leads to better overall performance, as the scanned mesh does not correspond well to the actual scene. R2 demonstrates that scene awareness further improves our result, where our agent learns important affordance information about chairs and sofas to improve the success rate. R3 highlights the importance of taking multiple gradient steps to obtain better gradients. When comparing R4 and R5 with R6, we can see that the learned function $\mathcal{G}$ is essential to obtain a better root translation and orientation, without which the agents drift significantly.

Table 3: Ablation study of different components of our framework.

| w/ Modified Scene | w/ Scene Awareness | MGP w/o multi-step | MPG w/o Geometric | MPG w/o TCN | SR ↑ | ACD ↓ | Accel. ↓ | Ground Freq. ↓ | Ground Pen. ↓ | Scene Freq. ↓ | Scene Pen. ↓ |
|---|---|---|---|---|---|---|---|---|---|---|---|
| ✗ | ✓ | ✓ | ✓ | ✓ | 95.0% | 252.1 | 9.8 | 0.0% | 26.4 | 0.0% | 18.4 |
| ✓ | ✗ | ✓ | ✓ | ✓ | 95% | 265.0 | 7.8 | 0.0% | 25.9 | 0.0% | 15.4 |
| ✓ | ✓ | ✗ | ✓ | ✓ | 96.7% | 326.9 | 8.6 | 0.0% | 24.8 | 0.0% | **11.4** |
| ✓ | ✓ | ✓ | ✗ | ✓ | 91.7% | 604.6 | 10.8 | 0.0% | 25.3 | 0.0% | 11.9 |
| ✓ | ✓ | ✓ | ✓ | ✗ | 95.0% | 223.2 | **7.8** | 0.0% | 26.6 | 0.0% | 18.5 |
| ✓ | ✓ | ✓ | ✓ | ✓ | 98.3% | **171.7** | 9.4 | **0.0%** | 24.9 | **0 %** | 16.6 |

## 5 Discussions

**Failure Cases.** We show the failure cases of our proposed method in the last two rows of Fig. 5. Although our method can estimate human pose and human-scene interaction from the PROX dataset with a high success rate, there is still much room for improvement. We simplify all humanoids and scene geometries as rigid bodies and do not model any soft materials such as human skin, cushions, or beds. As a result, our sitting and lying motion can often result in jittery motion, as contact occurs only in a very small surface area on the simulated body and hard surfaces. Due to the nature of physics simulation, our simulated humanoid sometimes gets stuck on narrow gaps between chairs and tables, or when lying on beds. In these scenarios, our method can often recover quickly by jolting out of the current state and trying to follow the target kinematic pose. Nevertheless, if the drift of global translation is severe or many 2D keypoints are occluded, our agent may lose balance. Our causal formulation may also lead to jittery motion when 2D keypoint observations jitter. Currently, our agent has no freedom to deviate from the 2D observations and always tries to match the 2D motion even when it is stuck. More intelligent control modeling will be needed to improve these scenarios.

**Conclusion and Future Work.** We introduce embodied scene-aware human pose estimation, where we formulate pose estimation in a known environment by controlling a simulated agent to follow 2D visual observations. Based on the 2D observation, its current body pose, and environmental features, our embodied agent acts in a simulated environment to estimate pose and interact with the environment. A multi-step projection gradient (MPG) is proposed to provide global movement cues to the agent by comparing the current body pose and target 2D observations. The experimental result shows that our method, trained only on synthetic datasets derived from popular motion datasets, can estimate accurate global 3D body pose on the challenging PROX dataset while using only RGB input. Future work includes providing a movement prior for the embodied agent to intelligently sample body motion based on scene features to better deal with occlusion.

**Acknowledgements:** This project is sponsored in part by the JST AIP Acceleration Research Grant (JPMJCR20U1).

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
