# Embodied Scene-aware Human Pose Estimation **Appendix**

**Zhengyi Luo**[1] * **Shun Iwase** [1]* **Ye Yuan**[1] **Kris Kitani**[1]

[1] Carnegie Mellon University

https://zhengyiluo.github.io/projects/embodied_pose/

## Summary

In this Appendix, we provide extended qualitative results, and more details about our Multi-step Projection Gradient as feature, evaluation, and implementation. **All code will be released for research purposes**. For supplementary videos, please refer to our project page.

## A Qualitative Results

### A.1 Supplementary Video

As motion is best seen in videos, we provide a supplemental video at our project site to showcase our method. Specifically, we provide qualitative results on the PROX and H36M datasets and also compare with state-of-the-art RGB-based methods (HuMoR-RGB and HyberIK). From the visual result, we can see that our method can estimate physically valid human motion from 2D keypoint inputs in a causal fashion and control a simulated character to interact with the scenes. Compared to non-physics-based method, ours demonstrates less penetration and remain relatively stable under

---

*Equal Contribution.

36th Conference on Neural Information Processing Systems (NeurIPS 2022).

occlusion. For future work, incorporating more physical prior and simulating soft objects such as cushions can further improve motion quality.

## B  Multi-step Projection Gradient (MPG)

In this section, we provide a detailed formulation for our multi-step projection gradient as a feature formulation. First, we provide our keypoint definition based on forward kinematics in Sec. B.1. Then, we will describe the geometric and learning-based transformation function $\mathcal{G}$ for augmenting MPG.

### B.1  Forward Kinematics and Keypoint Definition

The SMPL body model defines 24 body joints (green dots visualized in Fig. 1), which correspond to the 24 angles of the joints. For datasets such as H36M [5] and COCO [7], some joints of the SMPL human body do not have a one-to-one correspondence. Thus, prior SMPL-based human pose estimation methods often make use of a predefined joint regressor to recover the corresponding joints from the body vertices. As described in Sec.3.1, for the same body shape, the lengths of the limbs can change according to body pose, and changing the joint regressor may even increase performance [4]. However, it is computationally expensive to regress 3D joint locations with linear blend skinning at every MPG step. Thus, we opt to use a skeleton with fixed limb length and forward kinematics for our embodied agent toward real-time humanoid control. To obtain limb length, we use the rest pose (T-pose) of the SMPL body and recover the joint positions using the pre-defined joint regressor. From the joint positions, we compute the fixed limb lengths following the parent-child relationship defined in the kinematics tree.

Figure 1: Visualization of the 12 joints we use for pose estimation and the 2D keytpoins obtained from SMPL using forward kinematics. Red is the DEKR 2D keypoints detection result [2], white is our pose estimation result, and green (numbered) is the original 2D keypoints obtained from the projection joints of SMPL.

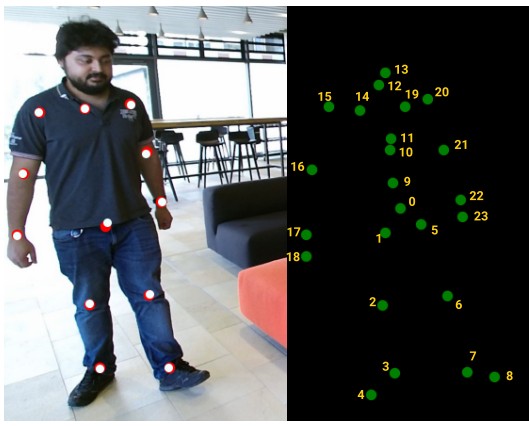

For both the COCO 2D keypoints [7] and H36M 3D joint positions of H36M [5], we use a subset of these points to obtain the corresponding points on the SMPL kinematics tree. Specifically, we use 12 joints, a similar joint definition to HuMoR [15], as visualized in Fig. 1. To obtain correspondences, we choose the following joint numbers from the 24 keypoints based on SMPL: #15, #16, #17, #20, #21, #22, #2, #3, #6, #7 without any processing. Then we average the position of joint #15 and #20 to obtain the mid-chest point. The pelvis point is obtained by averaging the 3 joints #0, #1, and #5. Following the process mentioned above, we obtain a sparse set of points (12) that we can directly obtain from the SMPL body pose parameters using forward kinematics to match visual observations and evaluation. Since not all 2D keypoints are confidently estimated, we use a confidence threshold of 0.1. For detected 2D keypoints of confidence $< 0.1$, we do not consider them and omit them during MPG calculation.

### B.2  MPG and Geometric and Learning-based Transform $\mathcal{G}$

We define the 3D body pose for time step $t$ as $\boldsymbol{q}_t \triangleq (\boldsymbol{r}_t^R, \boldsymbol{r}_t^T, \boldsymbol{\theta}_t)$ where $\boldsymbol{q}_t \in \mathcal{R}^{75}$. $\boldsymbol{r}_t^R \in \mathcal{R}^3$ and $\boldsymbol{\theta}_t \in \mathcal{R}^{69}$ are root and body joint orientations in axis-angles, while $\boldsymbol{r}_t^T \in \mathcal{R}^3$ is the global translation.

**Geometric Transform of $\mathcal{G}$ for $\partial r_t^R$.**  Due to depth ambiguity and severe occlusion, recovering global translation accurately from 2D observations alone is an ill-posed problem. Previous attempts that use image features[11] and 2D keypoints [12] can often result in unreliable translation estimates (as seen in Tab.2 and our quantitative video). Our pose gradient, on the other hand, provides *temporal* movement information instead of estimating global translation in one-shot. Using the temporal smoothness of human movement and the laws of physics, our pose gradient is more robust to

occlusion and will not produce sudden jumps in pose estimates. To provide better root translation, we also take advantage of a geometric transformation step to further improve $\partial r_t^{\mathrm{T}}$. Given 2D keypoints $\mathrm{kp}_{t+1}^{\mathrm{2D}}$ and body pose $q_t^k$ in this gradient descent iteration, we can treat a human skeleton as a rigid body to obtain a better translation estimate. Specifically, we compute the world-space 3D keypoints using forward kinematics $J_t = \mathrm{FK}_\beta(q_t^k)$ and optimize the following reprojection error:

$$\underset{\Delta r_t^{\mathrm{T}-k}}{\arg\min} \, ||\pi_{\mathrm{cam}}(J_t + \Delta r_t^{\mathrm{T}-k}) - \mathrm{kp}_{t+1}^{\mathrm{2D}}||_2^2 \tag{1}$$

We solve this equation using SVD to obtain $\Delta r_t^{\mathrm{T}\text{-}k}$ and $\partial r_t^{\mathrm{T}\text{-}k\prime}$ is updated by Eq. 2.

$$\partial r_t^{\mathrm{T}\text{-}k\prime} = \partial r_t^{\mathrm{T}\text{-}k} + \Delta r_t^{\mathrm{T}\text{-}k} \tag{2}$$

Notice that $\partial r_t^{\mathrm{T}\text{-}k\prime}$ is guaranteed to lead to a 2D reprojection error no bigger than the original $\partial r_t^{\mathrm{T}}$, so we use a direct additive transform to obtain a better global root translation gradient.

**Learning-based Transform of $\mathcal{G}$ for $\partial r_t^{\mathrm{T}}$.** Unlike global translation where depth ambiguity can cause sudden jumps in depth perception, we find that global rotation can be robustly estimated using a sequence of 2D keypoints only. Inspired by PoseTriplet [3], we use the temporal convolution model (TCN) [12] to recover the camera space 3D body orientation. Unlike PoseTriplet [3], we *only* regress the 3D body orientation and do not regress the 3D body pose: the TCN model is not quite robust to occluded or incorrect 2D keypoints (e.g. missing leg joints, which is prevalent when people are moving behind furniture). However, we notice that the TCN can still recover a reasonable root orientation by reasoning about reliable body joints (such as arms and torsos). Thus, we use the TCN to regress the camera space root orientation $r_t^{\prime \mathrm{R}} = \pi_{\mathrm{cam}}^{-1}(\mathcal{F}_{\mathrm{TCN}}(\mathrm{kp}_{t-\mu:t+1}^{\mathrm{2D}}))$ based on a window of 2D keypoints. We then calculate the difference between current root rotation $r_t^{\mathrm{R}}$ and $r_t^{\prime \mathrm{R}}$: $\Delta r_t^{\prime \mathrm{R}} = (r_t^{\mathrm{R}})^{-1} r_t^{\prime \mathrm{R}}$. We directly add $\Delta r_t^{\prime \mathrm{R}}$ as an additional entry to MPG instead of modifying $\partial r_t^{\mathrm{R}-k}$ since there is no guarantee that the 2D reprojection error from using $\Delta r_t^{\prime \mathrm{R}}$ will be better. We compute $\Delta r_t^{\prime \mathrm{R}}$ only once instead of K times. Combining the above two components, we obtain our complete MPG feature.

$$\phi_t = [\Delta r_t^{\mathrm{R}}, \partial r_t^{\mathrm{R}}, \partial r_t^{\mathrm{T}\text{-}k\prime}, \partial q_t] = \sum_{k=0}^{K} \mathcal{G}(\frac{\partial \mathcal{L}_{\mathrm{proj}}}{\partial q_t^k}, q_t^k, \mathrm{kp}_{t-\mu:t+1}^{\mathrm{2D}}) \tag{3}$$

## C  Details about Evaluation

### C.1  Evaluation Metrics

Here we provide a detailed list of our metric definition.

- SR: success rate measures whether the pose sequence has been estimated successfully. For optimization-based methods, LEMO [17] and HuMoR [15], it indicates whether the optimization process has successfully converged or not. For physics-based methods (ours), it indicates whether the agent has lost balance or not during pose estimation. We measure losing balance as when the position of the simulated agent $q_t$ and the output of the kinematic policy $\tilde{q}_t$ deviates too much from each other ($> 0.3$ meters per joint on average). For regression-based methods, *e.g.*, HyberIK [6], we do not use this metric.

- ACD (mm): one-way average chamfer distance measures the chamfer distance between the estimated human mesh and the point cloud captured from the RGB-D camera. This metric is a proxy measurement of how well the policy can estimate global body translation, as used in HuMoR [15]. Notice that RGB-D based methods such as HuMoR-RGBD and Prox-D both directly optimize this value, while our method does not use any depth information. Unstable optimization and degenerate solutions (where the humanoid flies away) can lead to a very large ACD value.

- Accel. (mm/frame$^2$): acceleration measures the joint acceleration and indicates whether the estimated motion is jittery. From the MoCap motion sequences from the AMASS dataset, a value of around $3 \sim 10$ mm/frame$^2$ indicates human motion that are smooth. When there is no ground-truth motion available, the joint acceleration serves as a proxy measurement for physical realism.

- Freq. (%): frequency of penetration measures how often a scene or ground penetration of more than 100 mm occurs. We obtain the scene penetration measurement by using the mesh vertices obtained from the SMPL body model.

- Pen (mm): penetration measures the mean absolute values of the penetrated vertices in the scene / ground, similar to the penetration metric in SimPOE [16].

- MPJPE (mm): mean per-joint position error is a popular pose-based metric that measures how well the estimated joint matches the ground-truth MoCap joint locations. This metric is calculated in a root-relative fashion where the root joint (#0, as shown in Fig.1 is aligned.

- PA-MPJPE (mm): mean per-joint position error after procrustes alignment.

- A-MPJPE (mm): absolute mean per-joint position error measures the absolute (non root-relative) joint position error in the world coordinate frame.

## C.2 Modified Scenes

As described in Sec.3.2, we make modified scenes based on the scanned meshes to better approximate the actual 3D scene. Due to the inaccuracy of scanning procedures and point cloud to mesh algorithms, the scanned meshes are often bulkier than the real environment. As a result, the gaps between chairs and tables are often too small in the mesh form, as visualized in Fig. 2. These inaccuracies can often cause the humanoid to get stuck between chairs and tables when trying to follow the 2D observations. To resolve this issue, we created modified scenes from the scanned meshes to better match the videos. We create these scenes semi-automatically by selecting keypoints in the mesh vertices and create rectangular cuboids and cylinders. Another advantage of using these modified scenes is that the signed distance functions for these modified geoms can be computed efficiently. Modified scenes are created for all scenes from the Prox and H36M datasets. All scenes are visualized in Fig.**??**.

Figure 2: Comparison between the scanned scene and our modified scene based on simple shapes. Notice that the scanned scene has much thicker chairs and tables, which leave smaller gaps for the agent to sit down.

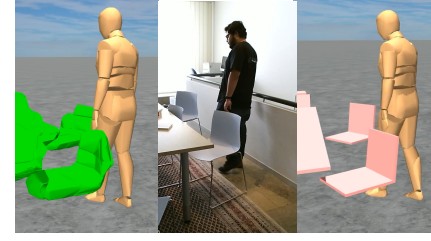

## C.3 Clarification about evaluation on H36M

We conduct quantitative evaluations on the H36M dataset, as shown in Table **??**. Since we do not make use of a joint regressor, we cannot perform the standard evaluation using the 14 joints commonly used for the evaluation on the H36M dataset. Instead, we use the 12 joints defined in Sec.B.1 to calculate MPJPE and A-MPJPE. As input, we use the same set of 12 2D keypoints. The 2D keypoints are estimated using a pre-trained DEKR [2] without additional fine-tuning on the H36M dataset. Our method only uses a sparse set of detected 2D keypoints (12 keypoints) as input and does not directly use any image feature. We also evaluate a different subset of 3D keypoints, as our use of forward kinematics $FK_\beta$ causes our joint definition to be different from prior works. For quantitative results, please refer to our supplementary video. This experiment also shows our method's ability to estimate pose in a generic setting where there are almost no scene elements (except for the ground and sporadically a chair).

# D Implementation Details

## D.1 Shape-based Universal Humanoid Controller

We extend the universal humanoid controller (UHC) proposed in Kin_poly [9] to also support humanoids with different body shapes. Specifically, we added the shape and gender parameter $\mathcal{H} \triangleq (\beta, \text{gender})$ to the state space of UHC to form our body shape conditioned UHC: $\pi_{\text{UHC}}(\boldsymbol{a}_t|\boldsymbol{s}_t, \widehat{\boldsymbol{q}}_t, \mathcal{H})$. In the original UHC, the body shape information is discarded during training, and the policy is learned using a humanoid based on the mean SMPL body shape. In ours, we create different humanoids for each body shape during training using the ground-truth body shape annotation from AMASS. In this

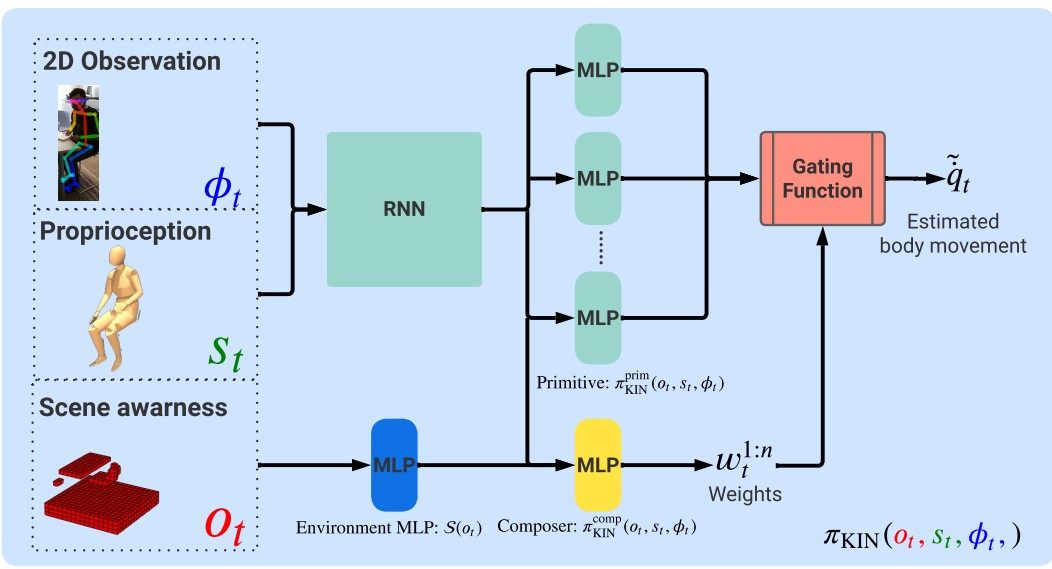

Figure 3: Network architecture for our scene-aware kinematic policy

way, we can learn a humanoid controller that can control humanoids with different body shapes. The action space and policy network architecture, and the reward function remain the same.

**Training procedure.** We train on the training split of the AMASS dataset [10] and remove motion sequences that involve human-object interactions (such as stepping on stairs or leaning on tables). This results in 11402 high-quality motion sequences. At the beginning of each episode, we sample a fixed-length sequence (maximum length 300 frames) to train our motion controller. Reference state initialization [14] is uesed to randomly choose the starting point for our sampled sequence and we terminate the simulation if the simulated character deviates too far from the reference motion $\hat{q}_t^p$. To choose which motion sequence to learn from, we employ a hard-negative mining technique based on the success rate of the sequence during training. For each motion sequence $\widehat{Q^i} = \hat{q}_0, \ldots, \hat{q}_T$, we maintain a history queue (of maximum length 50) for the most recent times the sequence is sampled: $h_1^i, \ldots h_{50}^i$ where each $h_0^i$ is a Boolean variable indicating whether the controller has successfully imitated the sequence during training. We compute the expected success rate $s^i$ of the sequence based on the exponentially weighted moving average using the history $s^i = \mathrm{ewma}(h_1^i, \ldots h_{50}^i)$. The probability of sampling sequence $i$ is then $P\left(\widehat{Q}^i\right) = \frac{\exp\left(-s^i/\tau\right)}{\sum_i^J \exp\left(-s^i/\tau\right)}$ where $\tau$ is a temperature parameter. Intuitively, the more we fail at imitating a sequence, the more likely we will sample it.

### D.2 Embodied Scene-aware Kinematic Policy

**Network Architecture and Hyperparameters** Our kinematic policy is an RNN-based policy with multiplicative compositional control [13]. It consists of a Gated Recurrent Unit (GRU)[1] based feature extractor to provide memory for our policy. Then, 6 primitives $\pi_{\mathrm{KIN}}^{\mathrm{prim}}$ are used to provide a diverse set of pose estimates based on 2D observation $\phi_i$ (MPG) and current body state $s_t$. We process the environmental features separately from the pose features due to its high dimensionality, using an MLP $\mathcal{S}_t$. The output of $\mathcal{S}_t$, concatenated with the RNN output, is fed into a composer $\pi_{\mathrm{KIN}}^{\mathrm{comp}}$

Table 1: Hyperparameters for our scene-aware kinematic policy.

|  | Batch Size | Learning Rate | Voxel grid | $w_{\mathrm{L1}}$ | $w_{\mathrm{prior}}$ |
|---|---|---|---|---|---|
| Value | 5000 | $5 \times 10^{-4}$ | $16 \times 16 \times 16$ | 5 | 0.0001 |
|  | TCN window | MPG - #steps | MLP-hidden | RNN-hidden | # primitives |
| Value | 81 | 5 | [1024, 512, 256] | 512 | 6 |

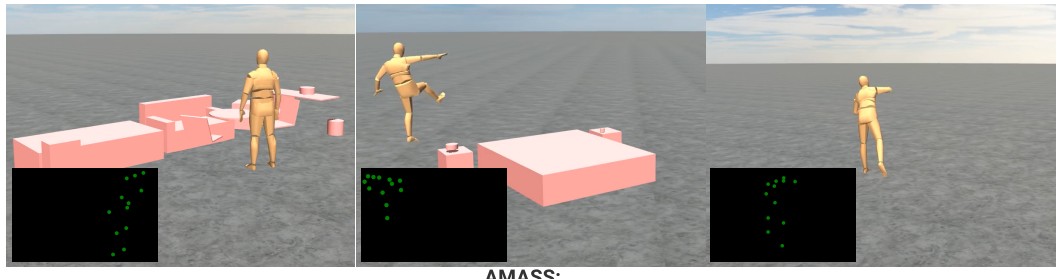

| AMASS: ACCAD-sway | AMASS: BioMotionLab_NTroje-kick | AMASS: BMLhandball-handball |

Figure 4: Visualization of a randomly sampled motion sequence and paired 2D keypoints for training. Here, we sample from the AMASS dataset (sequence name in the caption).

to produce the action weights. Intuitively, the primitive produces potential body movement based on the 2D observation and current body pose. Then, the composer chooses plausible action based on the environmental features. The detailed data flow of our policy is visualized in Fig. 3. The hyperparameters we used for training are shown in Table 1.

**Training procedure.** To train our kinematic policy, we sample motion sequences from AMASS [10], H36M [5], and Kin_poly [8] to generate synthetic paired 2D keypoints. To learn a network that is more robust to 2D keypoints detected from real videos, we randomly sample confidence from a uniform $[0, 1]$ distribution as the detection confidence. Keypoints with confidence $< 0.1$ will be omitted in our model. For each episode, we randomly sample a sequence, randomly change its heading, and sample a valid starting point in the camera frustum as visualized in Fig.**??**. We constrain the newly sampled sequence to be always visible in the image plane and re-sample if not. With a probability of 0.5, we also sample a random scene from the PROX dataset to let our agent learn human-scene interactions. Although the randomly sampled motion sequence does not always yield reasonable human-scene interactions (e.g. the reference motion may walk through a chair), collision and physics simulation will stop our agent automatically. In this way, our agent can learn to use the scene interaction term based on acting inside the scene.

# E  Broader Social Impact

This research focuses on estimating physically valid human poses from monocular videos. Such a method can be positively used for animation, telepresence, at home robotics, etc. where natural looking human pose can be served as the basis for higher-level methods. It can also lead to malicious use cases, such as illegal surveillance and video synthesis. As our method focuses on fast and causal inference, it is possible to significantly improve its speed and deploy it to real-time scenarios. Thus, it is essential to deploy these algorithms with care and make sure that the extracted human poses are with consent and not misused.