# OpenReview forum: "Embodied Scene-aware Human Pose Estimation"
_NeurIPS.cc/2022/Conference — NeurIPS 2022 Accept_

### Official Review · Reviewer_aaAC · 2022-07-08

**Rating:** 6
**Confidence:** 4
**Soundness:** 3 good
**Presentation:** 2 fair
**Contribution:** 2 fair

**Summary:**

This paper aims to perform single-camera pose estimation. It uses 2D key points and then uses a 3d simulated scene that the person is within, to enable an improvement in the pose estimation. The approach is tested in H36M and PROX datasets

**Questions:**

For the  MPG whatabout if temporal prediction neural networks such as LSTMs or transformers were used instead?
Could a further dataset demonstrate the performance better than H36M

**Limitations:**

none stated

**Strengths And Weaknesses:**

Strengths
The work doesn't require complex contact modelling or multi-stages of optimization processes, it is a single inference step.
The work proposes a temporal gradient projection to smooth the estimation over time.
The work is able to perform well on the POX dataset



Weaknesses
The H36M dataset is ill-suited for scene modelling as there are few objects within the foreground and seems to perform poorly and is distracting for the paper
The MPG is heavily based on the work of [3]
There is very limited experimental discussion around the work of the PROX dataset, making it hard for a reader to fully appreciate the performance of the approach
There is a disconnect between the 2D keypoint detection and the 3D/depth "perfect" generation of the scene,

---

> ### Author Response · Authors · 2022-08-02
> **Author Response to Reviewer aaAC (1/3)**
>
> Thank you so much for your feedback and suggestions. We are glad you found that our method "performs well on the PROX dataset". To address your comments and concerns:
>
> ---
>
> **Usage of H36M Dataset**
>
> We agree that the H36M dataset is not the best dataset for scene modeling: it only contains the ground and occasionally the chairs as scene elements. Thus, we mainly focus on demonstrating pose estimation results on the PROX dataset, which contains videos of humans interacting with a rich environment. We include the H36M dataset since it is a common benchmark for 3D human pose estimation, and we would like to establish a baseline from which we can compare with popular methods. It also shows that our method can be applied to global 3D human pose estimation, in both scene-rich and ground-only scenarios. We agree that more dataset that contains human scene interactions and ground-truth 3D pose annotation is needed to further research on this topic.
>
> As mentioned in Sec.4.1 and the caption for Table 1, our method only uses a sparse set of detected 2D keypoints as input and does not directly use image features. We also train only on synthetic 2D keypoints generated from motion datasets. As such, the comparison between ours and the SOTA methods is not a completely fair one. Methods such as VIBE, MeTRAbs, and HyberIK are all kinematics-only methods and have severe artifacts such as foot sliding, penetration, etc., as can be shown in the demo videos. SimPoE, though a physics-based method, uses VIBE as the first stage and can be viewed as a refinement of VIBE's result. Another important distinction is that we directly perform pose estimation in the **global** space (as compared to two-stage methods such as MeTRAbs and SimPoE, where the root position and body pose are separately regressed).
>
> For a better evaluation of the motion quality, we provide rendered videos of full sequences from the H36M dataset on our anonymous submission website [https://embodiedscene.github.io/embodiedpose/](https://embodiedscene.github.io/embodiedpose/). The qualitative result demonstrates that while our method may not achieve SOTA in joint positional error, it does produce a high-quality pose estimation.
>
>
> **MPG based on PoseTriplet [3]**
>
> Compared to PoseTriplet [3],  the only similarity is that we both make use of the network architecture from VideoPose [24], and our methodology and procedure are quite different. PoseTriplet uses the VideoPose network as the **main pose estimation backbone** and uses it to provide 3D joint positions for their physics-based imitator. Our MPG calculates the pose projection gradients through an iterative gradient descent process augmented with learning-based and geometric transform $\mathcal{G}$. The learning-based transform utilizes the VideoPose network to calculate the root orientation in camera space, while the geometric transform solves a rigid body transformation step using least-squares minimization. As such, we only utilize the VideoPose network in a **substep** of our MPG. The output is also different: MPG directly computes the pose gradient (in joint angles) in the global space instead of 3D joint positions in the camera space. The pose gradients serve as the movement cues for our velocity-predicting network. In Appendix B, we go into detail about how MPG is constructed using both learned and geometric transforms.

---

> > ### Author Response · Authors · 2022-08-02
> > **Author Response to Reviewer aaAC (2/3)**
> >
> >
> > **Discussion around the PROX Dataset**
> >
> > We are sorry for the lack of discussion around the PROX dataset and have added the following paragraph to the manuscript:
> >
> > "The PROX dataset contains RGB-D video sequences of humans interacting with indoor environments that are pre-scanned using depth cameras. We use the qualitative split of the PROX dataset, which contains 1,000K frames of videos of 20 subjects in 12 scenes. No ground-truth 2D keypoints or 3D poses are provided for this dataset. Due to the lack of training split, heavy occlusion, and human scene interactions, the PROX dataset is highly challenging for 3D human pose estimation methods, and popular methods often utilize multistage optimization to recover poses from videos."
> >
> > We also added a plot (Fig.4 in our revision manuscript) and the following paragraph for discussion of our result:
> >
> > "From Fig.4 we can see that our method can estimate accurate 3D absolute poses that align with the 3D scene. Compared to batch multistage optimization methods that utilize RGB-D input, PROX [7], and LEMO [46], our method can achieve comparable results while running in near real-time. Compared with the regression-based method, HybrIK [16] with RootNet [24], our method can estimate much more accurate root translation and body pose. In terms of physical plausibility, optimization-based methods often perform well but can still exhibit floating and penetration. While the optimization methods can take over a minute per frame to estimate poses, catastrophic failure can still occur (as shown in the fourth row). Since motion is best seen in videos, please refer to our demo videos [https://embodiedscene.github.io/embodiedpose/](https://embodiedscene.github.io/embodiedpose/) for quantitative comparison. "
> >
> > **Disconnect between 2D keypoint detection and 3D/depth "perfect" generation of the scene...**
> >
> > We are unsure of what exact disconnect is referred to here and would appreciate your understanding if we misinterpreted your meanings. Since the PROX dataset does not provide ground-truth 2D keypoints or 3D body pose annotation, we resort to off-the-shelf 2D keypoint detection methods, as done in prior arts [7, 31, 46]. The 3D scenes are also far from perfect (as demonstrated in Appendix C.2), with chair and sofa dimensions drastically different from their real-world counterparts. This creates a disconnect between the simulated environment and the real world that causes the agent to be stuck between chairs often. Thus, we resort to using modified scene geometries.
> >
> >
> > **Temporal prediction neural networks such as LSTMs or transformers**
> >
> > We indeed use a temporal prediction neural network in our network (please refer to Appendix D2). Our kinematic policy consists of Gated Recurrent Units (GRU) and processes the input autoregressively. We can not use a temporal prediction network alone (without MPG) since 2D keypoint observation is coupled with the camera pose. Given the same global body pose $\boldsymbol q_t$, different camera poses will produce different 2D keypoints. Since our network directly predicts the human pose in the global coordinate space, we cannot use 2D keypoints directly as input. The benefit of MPG is that it provides motion cues in the global coordinate space and can be directly used as network input. For more information and motivation of MPG, please refer to Sec.3.2.
> >
> >
> > **Limitations**
> >
> > Sorry for not making the limitations section more apparent. We discussed the failure cases of our methods in the "Sec.5 Discussion" and added additional failure cases in Appendix A.1. Here we can also provide a summary of the failure cases:
> >
> > Due to the discrepancy between physics simulation and the real world, our simulated humanoid can get stuck in narrow gaps between chairs, tables, and beds. Heavily erroneous 2D keypoint detection may also cause our humanoid to lose track and lose balance. Motions such as lying on the bed facing down and pushing up are also challenging since they are few examples in motion capture datasets. Although our method can often recover quickly by jolting out of the current state and trying to follow the target 2D keypoints, we can still observe failure cases where the humanoid cannot stay on track. For visualization of the failure cases (we include results on **all** sequences from the PROX dataset for evaluation), please refer to our anonymous website: [https://embodiedscene.github.io/embodiedpose/](https://embodiedscene.github.io/embodiedpose/).

---

> > > ### Author Response · Authors · 2022-08-02
> > > **Author Response to Reviewer aaAC (3/3)**
> > >
> > > **Further dataset demonstrate the performance better than H36M**
> > >
> > > Since we focus on pose estimation and human scene interaction modeling, we pick the PROX dataset as our main source of evaluation. We utilize the popular H36M dataset to establish a baseline and showcase that our method can estimate good-quality 3D human pose in a MoCap environment without ever being trained on real-world 2D keypoint detection (we train only on synthetic datasets). The same model is then used to evaluate on the PROX dataset, where we demonstrate the ability to estimate human pose and human scene interactions in a real-world environment from detected 2D keypoints. Our regression-based pose estimator has not been fine-tuned on the PROX dataset and has achieved SOTA results compared to batch-optimization-based methods [7, 31, 46].
> > >
> > >
> > > [7] M. Hassan, V. Choutas, D. Tzionas, and M. J. Black. Resolving 3D human pose ambiguities with 3D scene constraints. In International Conference on Computer Vision, Oct. 2019.
> > >
> > > [31] D. Rempe, T. Birdal, A. Hertzmann, J. Yang, S. Sridhar, and L. J. Guibas. Humor: 3d human motion model for robust pose estimation. In International Conference on Computer Vision441 (ICCV), 2021
> > >
> > > [40] Todorov, Emanuel et al. “MuJoCo: A physics engine for model-based control.” 2012 IEEE/RSJ International Conference on Intelligent Robots and Systems (2012): 5026-5033.
> > >
> > > [46] S. Zhang, Y. Zhang, F. Bogo, P. Marc, and S. Tang. Learning motion priors for 4d human body capture in 3d scenes. In International Conference on Computer Vision (ICCV), Oct. 2021

---

> > > > ### Comment · Reviewer_aaAC · 2022-08-09
> > > > **Reviewer response**
> > > >
> > > > Thank you for the author's responses, especially around the questions I had; I am significantly happier with the work and therefore will recommend a weak accept

---

> > > > > ### Author Response · Authors · 2022-08-09
> > > > > **Thank you for the response and update!**
> > > > >
> > > > > Thank you so much for the updated review and response! We are super glad that our reply has answered some of your concerns and improved your impression of our work. Feel free to ask us anything if there are any additional questions!
> > > > >
> > > > > Also, if we may ask, could you kindly update the ratings in the official review to "6: Weak Accept" and reflect the score change?
> > > > >
> > > > > Thanks again!
> > > > >
> > > > > Authors

---

> ### Author Response · Authors · 2022-08-08
> **Looking forward to your response!**
>
> Dear Reviewer aaAC,
>
>
> Thank you so much for all of your constructive feedback and suggestions. They will greatly help improve our work.
>
> We have provided comments on your concerns and revised our paper based on the suggestions. We would especially point to run-time and performance comparison with the SOTA methods both in the revised paper and [website](https://embodiedscene.github.io/embodiedpose/). We sincerely hope that our response could have addressed some of your concerns and improved your impression of our work.
>
> As the author's discussion period draws near an end, we would greatly appreciate it if you could acknowledge/update your initial reviews, and please do not hesitate if there are any more questions/concerns you would like to be addressed.
>
>
> Thanks again!
>
> Authors.

---

### Official Review · Reviewer_fkcV · 2022-07-11

**Rating:** 4
**Confidence:** 4
**Soundness:** 2 fair
**Presentation:** 3 good
**Contribution:** 2 fair

**Summary:**

The paper presents a 3d human pose framework based on both simulators and 2d/3d human estimators. It recovers both global 3D human poses and local human poses. The methods can be applied to daily activities such as PROX dataset.

**Questions:**

As I understand, the geometry in PROX is soft, is that reasonable to fully treat the sofas as hard/always flat? should we consider the deformation during the interaction of the geometries?

**Ethics Review Area:**

["I don’t know"]

**Limitations:**

Yes

**Strengths And Weaknesses:**

Strengths:
It sounds novel to me as I am not a simulator guy but a learning and 3d person guy. Using these simulators naturally will help generate lots of physics priors and encode the physics into the learning process.
I may not provide enough evidence for the novelty of the simulation part.

Weaknesses:
I did not see the comparison with "Neural MoCon: Neural Motion Control for Physically Plausible Human Motion Capture", if so, what is the difference, can you work on the dataset they are working on?

I think the performance is somewhat unsatisfying, we can see several metrics in Table 2 is not SOTA.

Several typos:
line 287:  scene penetration frequency Freq. and distance Pen..
line 204: keytpoints
line 202: rotationrR

---

> ### Author Response · Authors · 2022-08-02
> **Author Response to Reviewer fkcV (1/2)**
>
> The authors thank the reviewer for constructive and helpful feedback. We are glad you find our work "novel in utilizing simulation". To address your questions and concerns:
>
> ---
>
> **Comparison with Neural MoCon and Novelty**
>
> There are several differences between our method and Neural MoCon in methodology and approach.
>
> - Neural MoCon employs a **batch optimization based** reference motion initialization step to first obtain kinematic pose, while ours directly estimates physically valid 3D body pose in a causal and sequential manner. This two-stage approach renders Neural MoCon hard to employ in a streaming fashion. The two-stage formulation approach also creates a disconnect between the pose estimation stage and the later motion tracking stage. When the kinematic pose estimate is of lesser quality, the simulated motion tracker may find it hard to imitate the reference motion.
>
> - Neural MoCon employs a sampling-based control strategy, where the distribution prior to sample from is learned from the **training split** of the datasets. Since the GPA and H3.6M datasets all have similar actions between training and testing splits, it is natural that such a distribution prior is helpful in estimating motions from the test set. In the GPA dataset, only three sequences (0, 34, 52) out of 60 unique recordings were used for testing purposes, while the rest 57 are used for training. Ours, on the other hand, employs a Deep-RL based humanoid controller and does not utilize any additional distribution information from the testing datasets. As such, our method can be applied to the PROX dataset **without** ever being trained on it (the PROX data also do not provide any ground-truth 3D annotation).
>
> In all, compared to Neural MoCon, our method is casual, runs in near real-time, and does not need any training set to learn the prior data distribution. We also differ in philosophy, as our guiding principle is the idea of "embodiment" where we incorporate more egocentric features and scene awareness in body pose estimation. We use body perception, scene feature, and movement goal to control a humanoid to estimate pose in a streaming fashion, while Neural MoCon utilizes multiple stages of pose optimization and motion tracking.
>
> We can indeed work on the dataset they use (namely, GPA and GTA-IM). We chose PROX because, compared to the GPA dataset, it is more challenging and contains videos of humans interacting in a real-world indoor environment. The GPA dataset only contains videos recorded in a motion capture studio with subjects interacting with simple geometries such as stairs and cuboids. The background is also always a green screen. The PROX dataset has more heavy occlusion induced by real-world human-scene interactions (such as sitting on chairs and sofas), and the subjects do not wear any MoCap markers. Our method is able to perform well on the PROX dataset without ever training on the PROX videos. No additional modifications will be needed to run inference on the GPA dataset (as GPA also contains camera pose and scene geometries), though we choose the PROX dataset as it is more challenging and contains more realistic human-scene interactions.
>
> **Performance**
>
> We apologize for any confusion that our table might have caused. In Table 2, we compare with multistage batch-optimization-based methods such as HuMoR [31], PROX [7], and LEMO [46], and some of them use RGB-D sequences. The only regression-based method is HybrIk w/ RootNet, and it alone offers a fair comparison. Batch-optimization-based methods often directly optimize for the metrics shown in the table (such as ACD and scene penetration), which a regression-based method can only learn from data. We have added relative run-time for all of the methods in table 2 in our revision. We achieve SOTA results on ACD, ground, and scene penetration using RGB and about 100 times speedup while staying competitive with RGB-D results. We would also want to point out that the difficulty of the PROX dataset is the main reason why most of the methods that tackle this dataset is batch optimization based: factoring in the scene geometry, occlusion, and depth ambiguity in a regression-based method is exceedingly challenging. The lack of training data also exacerbates this issue. Here we include a subset of our main Table 2 where we only include the RGB-based methods:
>
> | Method (RGB)      | Phys. | Scene | SR $\uparrow$ | ACD $\downarrow$ | Time (seconds) $\downarrow$ |
> | ----------- | ----------- | ----------- | ----------- | ----------- | ----------- |
> | PROX      | ⤬  | ✔ | 100% | 297.81 | 47.64 |
> | HuMoR      | ⤬   | ⤬ | 98.3% | 435.7 | 11.73 |
> | HybrIK w/ RootNet      | ⤬  | ⤬  | - | 23512.7 | 0.08 |
> | Ours      |  ✔ | ✔ | 96.7% | 148.8 | 0.12 |
>
> From this table, where all the methods use RGB input, we can see that our method outperforms all baseline methods in terms of ACD (distance to point cloud) while having a per-frame runtime close to regression-based methods.

---

> > ### Author Response · Authors · 2022-08-02
> > **Author Response to Reviewer fkcV (2/2)**
> >
> > **Soft geometry in PROX**
> >
> > We thank the reviewer for the suggestion and question. Indeed, it would be beneficial to simulate soft bodies and make sofas and chairs more realistic. Our current simulation configuration is used to optimize run time and computational cost. While the Mujoco physics simulation does support soft body simulation, it is accomplished as a large link of spheres and capsules, which is computationally expensive to simulate. Memory consumption for contact modeling increases significantly with the number of geometries simulated, and it becomes impossible to simulate large scenes together with moving humanoids. As a result, soft body simulation is only used in small-scale tests such as robotic folding. We also do not have an annotation on the hardness and compliance of the sofas in the PROX scenes. As far as we know, we are the first work to simulate humanoids with large real-world scenes and we feel like it is a reasonable trade-off between realism and computational cost to start with rigid bodies. It is also an active research area to simulate interaction with deformable objects such as sofas and chairs.
> >
> >
> > **Typos and suggestions**
> >
> > We thank the reviewers for the careful review and reading. We have fixed the typos in the manuscript. Thanks!
> >
> > [7] M. Hassan, V. Choutas, D. Tzionas, and M. J. Black. Resolving 3D human pose ambiguities with 3D scene constraints. In International Conference on Computer Vision, Oct. 2019.
> >
> > [31] D. Rempe, T. Birdal, A. Hertzmann, J. Yang, S. Sridhar, and L. J. Guibas. Humor: 3d human
> > motion model for robust pose estimation. In International Conference on Computer Vision441
> > (ICCV), 2021
> >
> > [40] Todorov, Emanuel et al. “MuJoCo: A physics engine for model-based control.” 2012 IEEE/RSJ International Conference on Intelligent Robots and Systems (2012): 5026-5033.
> >
> > [46] S. Zhang, Y. Zhang, F. Bogo, P. Marc, and S. Tang. Learning motion priors for 4d human body capture in 3d scenes. In International Conference on Computer Vision (ICCV), Oct. 2021

---

> ### Author Response · Authors · 2022-08-08
> **Looking forward to your response!**
>
> Dear Reviewer fkcV,
>
>
> Thank you so much for all of your constructive feedback and suggestions. They will greatly help improve our work.
>
> We have provided comments on your concerns and revised our paper based on the suggestions. As the first work to simulate full-body humanoids in natural daily scenes, we not only provide improved performance on PROX (as can be shown in the revised paper and [website](https://embodiedscene.github.io/embodiedpose/)) but also much better run-time. We sincerely hope that our response could have addressed some of your concerns and improved your impression of our work.
>
> As the author's discussion period draws near an end, we would greatly appreciate it if you could acknowledge/update your initial reviews, and please do not hesitate if there are any more questions/concerns you would like to be addressed.
>
>
> Thanks again!
>
> Authors.

---

### Official Review · Reviewer_SrSt · 2022-07-12

**Rating:** 5
**Confidence:** 3
**Soundness:** 3 good
**Presentation:** 3 good
**Contribution:** 3 good

**Summary:**

The paper proposes a method for estimating 3D human pose given a monocular RGB video observing a human moving through a scene tha has been previously reconstructed (i.e. a scene for which a 3D mesh reconstruction is available).  The method is based on physical simulation of the human moving through a geometrically simplified version of the scene.  The simulation is driven by a multi-step projection gradient connecting 2D pose keypoints to the controller that drives the humanoid pose in simulation.

The experiments evaluate the proposed method as well as ablations against a breadth of methods from prior work on two datasets: PROX and H36M.  Performance is evaluated in terms of pose accuracy (mainly joint error metrics where ground truth pose is available) and pose plausibility (mainly distance to scene geometry, interpentretation frequency and distance etc.). The results show that the approach is competitive with prior work that uses image features on the H36M dataset, and mostly outperforms prior work on the PROX dataset.

**Questions:**

I would like the authors to respond to the point regarding scene simplification.

**Limitations:**

Limitations are described in the last section of the paper, mainly focusing on the simplification of human bodies and scene geometry to rigid bodies.  As far as I can tell, there is no discussion of potential negative societal impacts (despite a statement pointing to the supplement in the checklist).

**Strengths And Weaknesses:**

Strengths
+ The physics-based formulation is novel and offers a complemetary approach to the 3D human pose estimation problem statement relative to most prior work.  The fact that it can provide strong performance is quite impressive given the limited amount of input information that is given compared to other methods
+ The paper is well-written and presents a breadth of ablations that analyze the impact of different components of the method on performance

Weaknesses
- The approach relies on a "semi-automatic" simplification of the 3D scene geometry to address failure modes due to reconstruction noise.  I would have liked to see a more detailed discussion of this issue and also of the actual process for creating these simplified scenes.  This is an important detail as it constitutes a significant modification to the input available to the presented method relative to prior work.  Ideally, there would be an "ablation" that would report results based on the original (un-simplified) scene geometry to concretely measure the impact and value of this semi-automatic step

---

> ### Author Response · Authors · 2022-08-02
> **Author Response to Reviewer SrSt**
>
>
> We thank the reviewer for the positive, constructive, and helpful feedback. We are grateful that you find our work to have "strong performance" and "impressive given the limited amount of input". We also appreciate that you find our ablation to be comprehensive. To address your questions and concerns:
>
> ---
>
> **Modified Scene Geometries**
>
> We agree that modifying the geometries is an important part of our proposed method and will provide additional motivation in our revisions. In Appendix C.2, we provide visualizations that explain the motivations for this simplification step: due to inaccuracies of the mesh creation process, the scanned meshes provided by the PROX dataset often contain geometries bulkier than the original objects in the scene. As can be seen in Figure 3 of the Appendix, the chairs and tables are much thicker in the scanned mesh than in the real world. This makes it impossible for the humanoid to pass through. Since this issue does not fall under our pose estimation method,  we resort to using modified geometries to better reflect the real world. Another reason is that convex geometries are required for physics simulators [1]. Creating the convex hull of scanned meshes is a lossy process, and the resulting mesh often has unrealistic cracks and crevices.  As can be seen in Figure 3, a chair mesh must be decomposed into multiple segments of convex hulls to be faithfully simulated.
>
>
> Notice that our pose estimation networks itself **does not** require the modified geometries; it uses the signed distance function of the scene to query occupancy while the humanoid navigates the environments. The signed distance function can be acquired from a number of sources, such as scanned meshes or point clouds. Given better-formed geometries for physics simulation (e.g. convex hulls that are faithful to the original scenes), our method would require no additional modifications for utilizing the data. Advancement in physics simulation that alleviates the requirement of convex geometries for contact modeling may also help remove this geom modification process.
>
> For the semi-automatic scene modification process, given the scanned point cloud of the scenes, we first identify the objects that we wish to simplify. For each object, we manually select the geometries that we want to use to approximate it (e.g. cylinders and cuboids for tables). Then, for each segment of the objects (such as armrests and legs of chairs), we use a script to find the bounding box of the point clouds and calculate the parameters of the geometries (width and heights, diameters, etc.). Each scene is then represented by the compositions of these modified geometries.
>
> To better demonstrate the performance of our method without the scene simplification process, we rerun our method with the original scene scans and added an entry to our ablation table:
>
> | w/ Modified Scene     | w/Scene awareness |  MGP w/o multi-step | MGP w/o Geometric | MGP w/o TCN | SR $\uparrow$ | ACD $\downarrow$ | Accel $\downarrow$ |
> | ----------- | ----------- | ----------- | ----------- | ----------- | ----------- | ----------- | ----------- |
> |  ⤬  | ✔ | ✔ | ✔ | ✔ | 80% | 413.0 | 36.5 |
> |  ✔  | ✔ | ✔ | ✔ | ✔ | 96.7% | 148.2 | 9.2 |
>
> As can been seen in the table, without the scene modification process, the success rate drops while ACD increases. Notice that a few sequences (namely MPH11_00034_01 , MPH1Library_00145_01 , N0Sofa_00034_01) causes the simulation to become unstable and throws the following error:
>
> > Exception in do_simulation Got MuJoCo Warning: Nan, Inf or huge value in QACC at DOF 0. The simulation is unstable. Time = 54.2467. 1627
>
> Upon visual inspection, this is due to the humanoid getting stuck to the original mesh's unrealistic gaps and cracks and causes the simulation to become unstable. This does not happen in the modified scene, as it better reflects the real scene geometry.
>
>
>
> **Limitations**
>
> We thank the reviewer for feedback on the limitations and social impact. We apologize for this oversight and have added the following section in our appendix in revision:
>
> "This research focuses on estimating physically valid human poses from monocular videos. Such a method can be positively used for animation, telepresence, at home robotics, etc. where a natural-looking human pose can serve as the basis for higher-level methods. It can also lead to malicious use cases, such as illegal surveillance and video synthesis. As our method focuses on fast and causal inference, it is possible to significantly improve its speed and deploy it to real-time scenarios. Thus, it is essential to deploy these algorithms with care and make sure that the extracted human poses are with consent and not misused."

---

> ### Author Response · Authors · 2022-08-08
> **Looking forward to your response!**
>
> Dear Reviewer SrSt,
>
>
> Thank you so much for all of your constructive feedback and suggestions. They will greatly help improve our work.
>
> We have provided comments on your concerns and revised our paper based on the suggestions. As the first work to simulate full-body humanoids in natural daily scenes, we provide a viable and expandable approach (namely, modified scenes for better simulation) for this task. As simulation technology improves, we will be able to leverage better scene scans and simulators accordingly. We sincerely hope that our response could have addressed some of your concerns and improved your impression of our work.
>
> As the author's discussion period draws near an end, we would greatly appreciate it if you could acknowledge/update your initial reviews, and please do not hesitate if there are any more questions/concerns you would like to be addressed.
>
>
> Thanks again!
>
> Authors.

---

### Official Review · Reviewer_wM8A · 2022-07-12

**Rating:** 8
**Confidence:** 4
**Soundness:** 4 excellent
**Presentation:** 4 excellent
**Contribution:** 4 excellent

**Summary:**

In this paper, the authors propose a scene-aware pose estimation framework based on the concept of embodiment. Different from previous work with multistage optimization, non-causal inference, and complex contact modeling for the scene-aware pose, this method is simple(only one stage) and recovers robust scene-aware pose in the simulated environment. Besides, to simulate good poses, they disentangle the camera pose and use a multi-step projection gradient defined in the global coordinate frame as the movement cue for our embodied agent.
This method only needs 2D observation and can be trained on synthetic datasets for real-world pose estimation. They achieve the state-of-the-art on PROX dataset without using any training data in this dataset.

**Questions:**

1) It would be better to discuss how the contact information models in this paper. I think it is one of the most important differences between this work and other simulation-based mocap methods.
2) Does this method need to train new agents for different motion sequences?

**Ethics Review Area:**

["I don’t know"]

**Limitations:**

Please follow the weakness.

**Strengths And Weaknesses:**

Strengths:
+ The main contribution--simulating scene-aware humanoid to mimic visual observation and recover 3D poses in the real-world environment is significant. I believe it will be a good baseline for the scene-aware pose estimation.
+ This method achieves promising results on the challenging poses of PROX dataset.
+ The MGP is reasonable on 3D pose recovery and achieves significant improvements shown in ablation studies.
+ This paper is well written and easy to follow.
Weakness:
-The simplified scene representation causes some failure cases in the poses of sitting or laying.
- The contact modeling between motion and scene is a little bit unclear.

---

> ### Author Response · Authors · 2022-08-02
> **Author Response to Reviewer wM8A**
>
> We thank the reviewer for the positive, constructive, and encouraging review. We are super glad that you find our work "simple and robust", "significant", our result on PROX "state-of-the-art", and our method "only needs 2D observations and can be trained on synthetic dataset for real-world pose estimation". To address your questions and concerns:
>
>
> **Failure cases in the poses of sitting or lying**
>
> We agree that the performance for some of the lying and occluded sitting sequences still has room for improvement. As discussed in failure cases (Sec.5 and Appendix A.1), the humanoid and scene geometries do not perfectly reflect the real-world and its physical properties. Beds and sofas are extra challenging due to their soft cover and pliable surfaces. Although physics simulation such as Mujoco supports simulating soft bodies (through simulating hundreds of small spheres and capsules), it is extremely computationally expensive (both in CPU and memory consumption) to simulate a larger amount of soft bodies. Thus, in this work, we choose to use modified geometries to better replicate the scenes. We wish to be transparent about the failure cases and shortcomings of our method and have included all of the available sequences from the PROX dataset on our anonymous website: [https://embodiedscene.github.io/embodiedpose/](https://embodiedscene.github.io/embodiedpose/).
>
>
>
> **Contact modeling**
>
>
>  Our learning framework models contact through the input pose $\mathbf{q_t}$ and environmental sensors. Unlike previous simulation-based Mocap methods such as SimPOE [2], our pose estimation network takes into account the current simulated pose $\mathbf{q_t}$ into consideration while producing its pose estimation $\tilde{\mathbf{q_t}}$. This closes the loop between physics simulation and pose estimation, where the network is aware of the current simulation states while producing the next time-step estimates. Prior arts, like SimPoE, often use a two-stage approach where the kinematic pose is estimated independently from the physics simulation process. As a result, our network receives real-time feedback of contact and collision from the physics simulation through $\mathbf{q_t}$, as the reaction in $\mathbf{q_t}$ indicates that the humanoid cannot walk through certain obstacles. Combined with environmental occupancy information, our network can learn the association between scene occupancy and collision through a large number of simulation experiences.
>
>
> [1] Todorov, Emanuel et al. “MuJoCo: A physics engine for model-based control.” 2012 IEEE/RSJ International Conference on Intelligent Robots and Systems (2012): 5026-5033.
>
> [2] Yuan, Ye et al. “SimPoE: Simulated Character Control for 3D Human Pose Estimation.” 2021 IEEE/CVF Conference on Computer Vision and Pattern Recognition (CVPR) (2021): 7155-7165.
>
>
> **New agents for different motion sequences**
>
> We do not require new agents trained for different motion sequences. Our evaluation on the PROX and H36M datasets is carried out without ever using any motion sequences or fine-tuning our model, and a single policy is used to evaluate all sequences from the PROX and H36M datasets. Once learned, our agent can estimate poses of everyday activities without additional information or training. It is conceivable and possible that specialized agents can be trained for more dynamic motion sequences such as competitive sports for better performance.

---

> ### Author Response · Authors · 2022-08-08
> **Looking forward to your response!**
>
> Dear Reviewer wM8A,
>
>
> Thank you so much for all of your constructive feedback and suggestions. They will greatly help improve our work.
>
> We have provided comments on your concerns and revised our paper based on the suggestions. Since our method is causal, runs in sub-real time, and can be applied with and without a real-world scene, it is not restricted to the use cases demonstrated in the paper (we have included a teaser in-the-wild result in the anonymous [webiste](https://embodiedscene.github.io/embodiedpose/), which uses the same model reported in the paper. We sincerely hope that our response could have addressed some of your concerns and improved your impression of our work.
>
> As the author's discussion period draws near an end, we would greatly appreciate it if you could acknowledge/update your initial reviews, and please do not hesitate if there are any more questions/concerns you would like to be addressed.
>
>
> Thanks again!
>
> Authors

---

### Author Response · Authors · 2022-08-03
**General responses and revision summary**

We thank all reviewers for their time and constructive feedback and hope that our responses provide clarification on our approach and results. Here we summarize some common clarification points and provide a list of revisions we have made based on the constructive feedback.

**Performance**

In this work, we explore pose estimation using "embodiment" as our guiding principle, where we incorporate egocentric features and scene awareness in our pose estimation pipeline. We train a simulated agent to follow 2D keypoint observations and navigate in recreated real-life environments. Our method is trained on **motion sequences** from the AMASS, kin_poly, and H36M (train split) datasets and does not utilize any image-level features from the video sequences. At inference time, we used an off-the-shelf 2D keypoint detector to extract 12 2D keypoints compatible with the SMPL humanoid body (more details in Appendix B.2) and use them as input in a streaming fashion. In this constrained setting, our model is able to achieve the state-of-the-art result on the challenging PROX dataset without ever seeing the videos or 2D keypoint observations from PROX during training. Compared to SOTA batch optimization-based methods, our approach can run in a fraction of the runtime while remaining competitive in performance. As motion is best seen in videos, we provide visualization of all the sequences from the PROX dataset on our [anonymous website](https://embodiedscene.github.io/embodiedpose/).


**Modified Scene Geoms**

We utilize modified scene geometries to better approximate the real world, as early in our development we realize that the convex decomposition of the scanned scenes is a poor substitute for the real scenes. As can be shown in Appendix C.2, the scanned scenes are often bulkier than their real-world counterparts and contain cracks and crevices. This is a result of both inaccuracies in the scene scanning process and the convex decomposition necessary to port those scenes into a physics simulation. Thus, we modify the scenes and use simple geometries to better approximate their real-world counterparts, and offer smoother and stabler shapes for physics simulation. We believe that as simulation and 3D scanning technologies progress, as a community, we will be able to create better digital copies of real scenes. As one of the first works to simulate humanoids together with real-life-sized scenes, we modify the scenes so that it offers the best trade-off between simulation speed and realism. Notice that our core contribution, our MPG formulation, and humanoid control methods, do not depend on this operation. The only input it requires is the signed distance function (sdf) of the scene to query occupancy, and the sdf can come from a number of sources including point clouds and meshes. As simulation technology progresses, our method can easily be adapted to leverage better-approximated scenes.


**List of Revisions**

1. Fixes in Table 1 and Table 2, where the checkmarks for HyberIk and PROX were erroneous. HyberIk does utilize the image features on the H36M dataset. PROX-RGB also utilizes the scene features during their optimization procedure.
2. Added the runtime in seconds to each of the methods shown in Table 2. We hope this puts the performance of our method in better perspective, especially compared to multi-stage batch optimization-based methods.
3. Added ablation study on not using the modified scenes to our ablation study.
4. Added the discussion of the experimental result of the PROX dataset as well as Fig. 4 for qualitative evaluation.
5. We change the notation of scene features from $\boldsymbol s_t$ to $\boldsymbol o_t$, since $\boldsymbol s_t$  is already used to refer to simulation states.
7. Appended discussion of social impact in the Appendix.
6. Typo fixes

We appreciate all suggestions for our papers and believe that they have made our manuscript stronger. Please let us know if there is any remaining questions or clarifications we could make. Thanks!

---

### Meta-Review · Area_Chair_Mmbv · 2022-08-24

**Recommendation:** Accept
**Confidence:** Certain

**Metareview:**

The submission initially received mixed reviews.  After rebuttal, all reviewers felt their concerns reasonably addressed and recommended acceptance (though one didn't update the score).  The AC agrees. The authors are encouraged to revise the paper accordingly.

**Award:**

No

---

### Decision · Program_Chairs · 2022-09-14

Accept